# Ubiquitination drives COPI priming and Golgi SNARE localization

**Swapneeta S Date[1], Peng Xu[1], Nathaniel L Hepowit[2], Nicholas S Diab[1], Jordan Best[1], Boyang Xie[1], Jiale Du[3], Eric R Strieter[3], Lauren P Jackson[1], Jason A MacGurn[2], Todd R Graham[1]\***

[1]Department of Biological Sciences, Vanderbilt University, Nashville, United States; [2]Department of Cell and Developmental Biology, Vanderbilt University, Nashville, United States; [3]Department of Chemistry, University of Massachusetts Amherst, Amherst, United States

**Abstract** Deciphering mechanisms controlling SNARE localization within the Golgi complex is crucial to understanding protein trafficking patterns within the secretory pathway. SNAREs are also thought to prime coatomer protein I (COPI) assembly to ensure incorporation of these essential cargoes into vesicles, but the regulation of these events is poorly understood. Here, we report roles for ubiquitin recognition by COPI in SNARE trafficking and in stabilizing interactions between Arf, COPI, and Golgi SNAREs in *Saccharomyces cerevisiae*. The ability of COPI to bind ubiquitin, but not the dilysine motif, through its N-terminal WD repeat domain of β'-COP or through an unrelated ubiquitin-binding domain is essential for the proper localization of Golgi SNAREs Bet1 and Gos1. We find that COPI, the ArfGAP Glo3, and multiple Golgi SNAREs are ubiquitinated. Notably, the binding of Arf and COPI to Gos1 is markedly enhanced by ubiquitination of these components. Glo3 is proposed to prime COPI–SNARE interactions; however, Glo3 is not enriched in the ubiquitin-stabilized SNARE–Arf–COPI complex but is instead enriched with COPI complexes that lack SNAREs. These results support a new model for how posttranslational modifications drive COPI priming events crucial for Golgi SNARE localization.

**\*For correspondence:**
tr.graham@vanderbilt.edu

**Competing interest:** The authors declare that no competing interests exist.

## Editor's evaluation

This article will be of interest for cell biologists focused on understanding membrane biology, trafficking, and protein ubiquitination, as well as yeast geneticists. The main finding of this paper is that non-degradative ubiquitination is an important mechanism driving COPI-dependent SNARE trafficking and localization.

## Introduction

The sorting of proteins in the endomembrane system is a highly regulated, vesicle-mediated process important for many physiological events. Coat proteins drive the formation of vesicles by assembling onto the cytosolic surface of cellular membranes, where they select cargo proteins (***Bonifacino and Glick, 2004***; ***Brandizzi and Barlowe, 2013***; ***Schmid, 1997***; ***Spang, 2013***). COPI-coated vesicles originate at the Golgi or endosomes, and mediate retrograde transport to early Golgi cisternae or back to the endoplasmic reticulum (ER) (***Bonifacino and Glick, 2004***; ***Letourneur et al., 1994***; ***Tojima et al., 2019***). COPI is a highly conserved heptameric protein complex (α, β, β', γ, δ, ε, and ζ subunits) that is recruited to Golgi membranes by the small GTP-binding protein Arf (Arf1 and Arf2 in budding yeast) (***Hsu, 2011***; ***Serafini et al., 1991***; ***Thomas and Fromme, 2020***; ***Waters et al., 1991***). The N-terminal WD repeat (WDR) domains of α- and β'-COP recognize sorting signals on cargoes, such

as dilysine motifs (KKxx and KxKxx) commonly found at the C-terminus of ER-resident membrane proteins (*Eugster et al., 2004*; *Jackson et al., 2012*; *Ma and Goldberg, 2013*). Mutations that impair the ability of both α- and β′-COP to bind dilysine motifs disrupt trafficking of well-studied dilysine cargoes, but these mutations do not affect cell viability (*Jackson et al., 2012*). However, COPI is essential in yeast and cells are unviable when both terminal propeller domains of α- and β′-COP are deleted (*Eugster et al., 2004*). These observations imply that important but poorly understood mechanisms exist for COPI recognition of cargoes lacking dilysine motifs. As Golgi cisternae mature from *cis* to *trans* in budding yeast, the retrograde movement of resident proteins, which primarily lack canonical dilysine signals, becomes critical in order to maintain a functional Golgi. Resident Golgi proteins, thus, are also important COPI cargo (*Banfield, 2011*; *Glick et al., 1997*; *Ishii et al., 2016*; *Kurokawa et al., 2019*).

SNAREs are another critical cargo of COPI vesicles because they are essential for vesicle fusion with the target membrane and are proposed to prime, or nucleate, coat formation (*Chen and Scheller, 2001*; *Rein et al., 2002*; *Rothman and Warren, 1994*; *Springer et al., 1999*). In addition to incorporating v-SNAREs into newly forming vesicle membranes, COPI must also mediate retrograde transport of early Golgi t-SNAREs moving through the Golgi by cisternal maturation to maintain Golgi organization, but how COPI mediates sorting of SNAREs is poorly understood. Because of the tail-anchored topology of SNAREs, none of these proteins contain a C-terminal dilysine motif on the cytosolic side of the membrane where it is accessible to COPI. Few sorting signals have been identified in SNARE proteins and how they are incorporated into COPI vesicles is incompletely understood (*Black and Pelham, 2000*; *Daste et al., 2013*; *Fukasawa et al., 2004*; *Gao and Banfield, 2020*; *Jackson et al., 2010*; *Mancias and Goldberg, 2007*; *Martinez-Arca et al., 2003*; *Miller et al., 2007*). The ArfGAP protein Glo3 may contribute to SNARE localization because it is known to interact with COPI, Arf-GTP, and SNAREs and is proposed to be part of the priming complex (*Rein et al., 2002*). In addition, Glo3 stimulates conversion of Arf-GTP to Arf-GDP, which destabilizes the COPI coat and is crucial for vesicle uncoating (*Tanigawa et al., 1993*). How these Glo3 interactions are regulated to keep the Arf-GAP activity of Glo3 in check to allow Arf-GTP-mediated COPI assembly during vesicle formation is unclear.

We recently found that COPI plays a role in the recycling of a budding yeast v-SNARE, Snc1, from the endocytic pathway to the *trans*-Golgi network (TGN) through recognition of a polyubiquitin (polyUb) signal (*Xu et al., 2017*). The cargo-binding WDR domains of α-COP and β′-COP bind specifically to polyUb (*Xu et al., 2017*). Deletion of the β′-COP N-terminal WDR domain (β′-COP Δ2–304) disrupts Snc1 recycling while the replacement of this domain with unrelated ubiquitin-binding domains (UBDs) restores Snc1 recycling. In addition, the β′-COP Δ2–304 mutant displays a slow growth phenotype, which is fully corrected by addition of the unrelated UBD. Because the β′-COP–UBD fusion proteins are incapable of binding dilysine motifs, these observations suggest a direct role for COPI–Ub interaction in Snc1 recycling (*Xu et al., 2017*). Thus, β′-COP plays a critical role in this ubiquitin-dependent trafficking route, but it was unclear if the COPI–ubiquitin interaction is important for trafficking of any other cargoes. The only other cargo known to be missorted in the β′-COP Δ2–304 mutant is Emp47, which bears a variant dilysine motif (KxKxx) that is uniquely recognized by the β′ propeller domain and the trafficking of Emp47 does not require ubiquitin interaction (*Eugster et al., 2004*; *Xu et al., 2017*). Another COPI cargo analyzed, Rer1-GFP, is sorted normally by β′-COP Δ2–304 (*Xu et al., 2017*).

In the current study, we seek to determine if the COPI–ubiquitin interaction is a general principle of SNARE trafficking and to define the molecular interplay between ubiquitination, COPI, and COPI vesicle components including Arf, ArfGAP, and COPI cargo. We show that the normal localization of several SNAREs functioning at the ER–Golgi interface or within the Golgi, including Bet1, Gos1, Snc2, Bos1, and Sec22, requires COPI–ubiquitin interactions. In addition, several Golgi SNAREs, COPI subunits, and the ArfGAP Glo3 are ubiquitinated under physiological conditions in *Saccharomyces cerevisiae*. Importantly, we show that ubiquitination of these components strengthens the interaction between Golgi SNAREs and COPI while apparently excluding Glo3, providing critical new mechanistic insight into priming mechanisms for COPI vesicle formation. These studies highlight the finely orchestrated role of ubiquitination in driving COPI priming and sorting of a specific set of Golgi SNAREs crucial to the functional organization of Golgi.

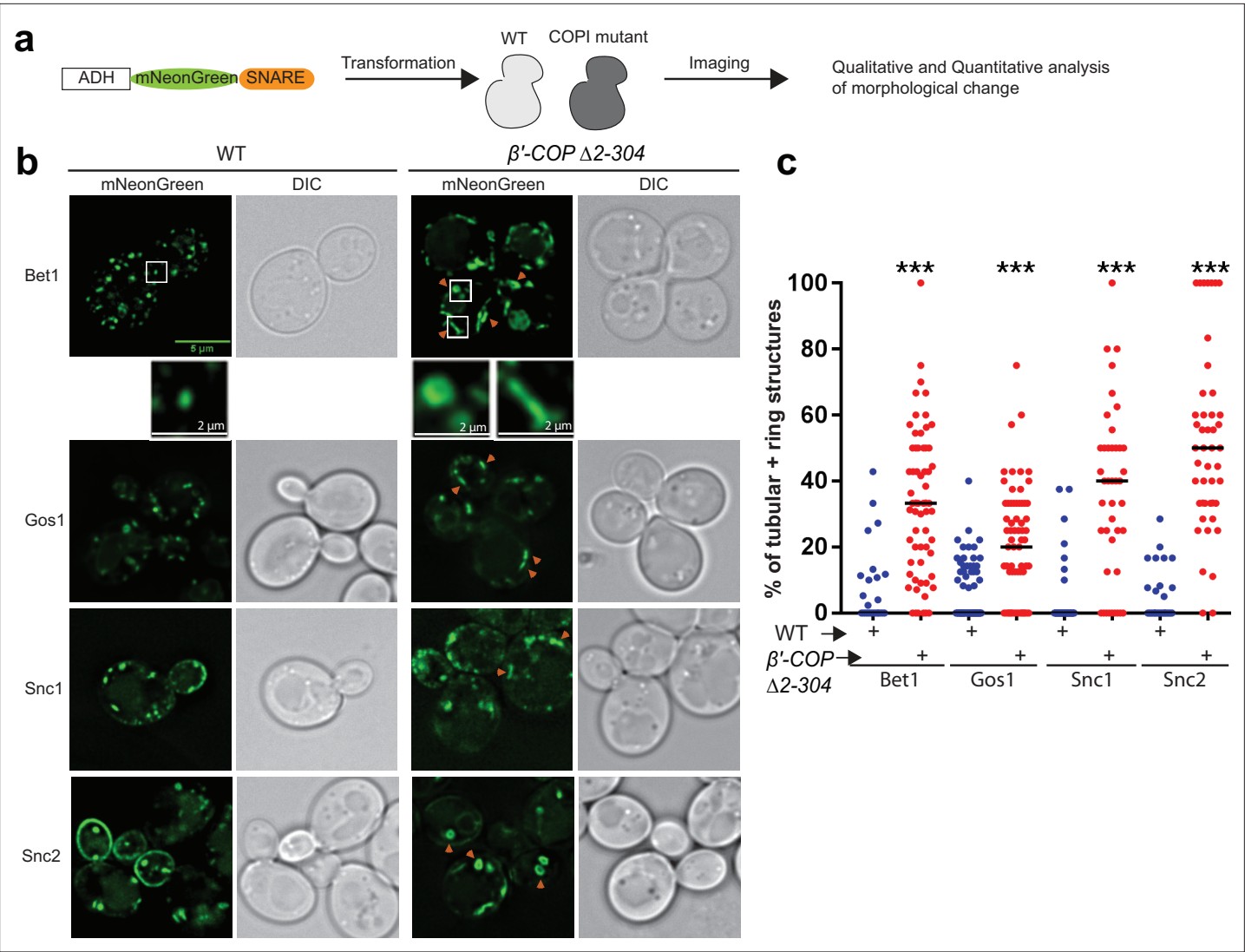

**Figure 1.** SNARE localization to morphologically aberrant structures in COPI mutants. (**a**) Schematics of the experimental setup wherein SNAREs are individually tagged with mNeonGreen and expressed under constitutive ADH promoter in *Saccharomyces cerevisiae* wild-type (WT) cells or in cells with a deleted N-terminal WDR of β'-COP (Δ2–304). (**b, c**) Live-cell imaging shows significant differences in the appearance of SNAREs Bet1, Gos1, Snc1, and Snc2 where elongated tube- and ring-like structures (orange arrowheads) are seen in β'-COP Δ2–304 cells compared to control cells with full-length β'-COP (WT). Tubes were defined by having a long axis (length) at least twice as long as the short axis. The rings were defined as spherical structures having a larger diameter than a normal puncta and with a dimly fluorescent center. Data in (**c**) are the percentage of structures per cell optical section that are rings and tubules. The remainder of the fluorescent structures are punctae. Statistical differences were determined using a one-way analysis of variance (ANOVA) on the means of the three biological replicates (***p < 0.001). Scale bars represent 5 µm in the full panels and 2 µm for the insets. Full panel images are all scaled equally.

The online version of this article includes the following figure supplement(s) for figure 1:

**Figure supplement 1.** Localization of six SNAREs is perturbed in β'-COP Δ2–304 mutant.

**Figure supplement 2.** Bet1, Snc1, and Snc2 localization to aberrant membranes in β'-COP Δ2–304 mutants is independent of expression level.

## Results

### A subset of SNAREs is mislocalized in the β'-COP Δ2–304 mutant

To determine the dependence of SNARE localization on COPI–ubiquitin interactions, we individually tagged 17 yeast SNAREs with mNeonGreen (mNG) and expressed them in *S. cerevisiae* wild-type (WT) cells or in a strain where the ubiquitin-binding N-terminal WDR of β'-COP had been deleted (β'-COP Δ2–304) (*Figure 1a* and *Supplementary file 1*, Table 1; *Xu et al., 2017*). This β'-COP mutation does not completely eliminate COPI polyUb binding because α-COP can also bind polyUb (*Xu*

*et al., 2017*) therefore, the SNAREs were overexpressed from a strong ADH promoter so the screen would be more sensitive for detecting changes in localization. Many of the mNG-SNARE fluorescent patterns were indistinguishable between WT and β'-COP Δ2–304 cells (*Figure 1—figure supplement 1a*). However, for Bet1, Gos1, Snc1, and Snc2, a significant accumulation of individual SNAREs to elongated tube- and ring-like structures was observed in β'-COP Δ2–304 (*Figure 1b, c*). Bet1 and Snc2 were found in ring- and tube-shaped structures while Gos1 was rarely observed in the rings. As previously shown for Snc1 (*Xu et al., 2017*), Snc2 plasma membrane localization was also reduced in the COPI mutant. Tlg1 also localized to enlarged tubular structures and Snc1 colocalizes to these enlarged structures in the β'-COP Δ2–304 cells (*Figure 1—figure supplement 1a*; *Xu et al., 2017*). In addition, Sec22 and Bos1 were partially mislocalized to vacuolar structures in β'-COP Δ2–304 cells (*Figure 1—figure supplement 1b*). GFP is typically cleaved from protein chimeras upon arrival in the vacuole. Consistently, immunoblotting of cell lysates with anti-GFP indicated that 40% of the GFP-Sec22 chimera was cleaved in β'-COP Δ2–304 cells to release free GFP, in contrast to WT cells where less than 5% was cleaved (*Figure 1—figure supplement 1c, d*).

To test whether the observed morphological changes were caused by SNARE overexpression or loss of the β'-COP WDR domain, we expressed the Bet1, Snc1, and Snc2 mNG constructs using the weaker, inducible *CUP1* promoter with a short (1 hr) induction time. Short induction times using the *CUP1* promoter have been shown to approximate physiological protein abundance for SNAREs Bet1 and Snc1 (*Best et al., 2020*). Comparable morphological changes were observed with these SNAREs localizing to elongated tubular- and ring-like structures in β'-COP Δ2–304 relative to WT cells (*Figure 1—figure supplement 2a, b*). Thus, Bet1, Snc1, and Snc2 were localized to aberrant structures whether they were expressed using a strong, constitutive ADH promoter or the weaker, inducible *CUP1* promoter. All subsequent imaging studies used the *CUP1* promoter to approximate physiological SNARE expression. Together, these data suggest a dependence of a subset of SNAREs on COPI–ubiquitin interactions for their proper localization.

To further characterize the morphological changes observed for SNAREs, we performed colocalization analysis of mNG-Gos1 and mNG-Bet with four Golgi markers (early Golgi markers, Anp1 and Mnn9 and late Golgi markers Sec7 and Chs5) (*Zhu et al., 2019*). mNG-Gos1 colocalized substantially with Chs5, Sec7, and Mnn9 in WT cells but showed a significant decrease in colocalization with these markers in in β'-COP Δ2–304 cells. Gos1 weakly colocalized with Anp1 in WT cells and this overlap was also diminished in β'-COP Δ2–304 (*Figure 2a, b*). mNG-Bet1 also showed a significant reduction in colocalization with all four Golgi markers in β'-COP Δ2–304 (*Figure 2—figure supplement 1a, b*). We also observed colocalization of Bet1 with the medial-Golgi marker Aur1 in WT cells and the degree of this colocalization was also reduced in β'-COP Δ2–304 (*Figure 2—figure supplement 2a, b*). Where colocalization was observed between mNG-SNAREs and Golgi markers in β'-COP Δ2–304 cells, the Golgi marker only overlapped with a small region of the tube-like structures labeled with the mNG-SNARE (*Figure 2a*, *Figure 2—figure supplement 1*). We suspect these incidents of colocalization are adjacent structures, but it is possible they are single compartments where the two proteins are mostly segregated into separate domains. Thus, there was loss of mNG-Gos1 and mNG-Bet1 from the Golgi compartments where they normally localize when the terminal UBD of β'-COP was deleted. In addition, there was no noticeable difference in Anp1, Mnn9, Chs5, or Sec7 localization or the morphology of the Golgi compartments these proteins marked. Similarly, no morphological difference for early- and medial-Golgi markers, Sed5 and Aur1, respectively, was observed for β'-COP Δ2–304 cells compared to WT cells (*Figure 2—figure supplement 3*).

To further characterize the Gos1-labeled structures and Golgi morphology, we calculated the abundance of intracellular fluorescent structures grouped as either punctate structures (approximately 0.45 µm dotted structures), or tube-like structures (structures for which the longer axis is 2–3× longer than the shorter axis, length of the longer axis ranges from 0.6 to 1.8 µm) and ring-like structures (enlarged punctate-type structures but with a hole in the middle akin to a ring, approximate size 0.6–0.9 µm across both long and short axis) (*Figure 2c*). No difference between WT and β'-COP Δ2–304 was observed for the total number of mNG-Gos1 or mNG-Bet1 fluorescent structures per optical section. However, relative to WT cells, a greater number of these structures were rings and tubes in the β'-COP Δ2–304 cells with a concomitant reduction in punctae. For the Golgi markers, the number of punctae per cell optical section was the same in WT and mutant cells and the tube or ring structures were rarely observed. The average size of intracellular structures was not significantly

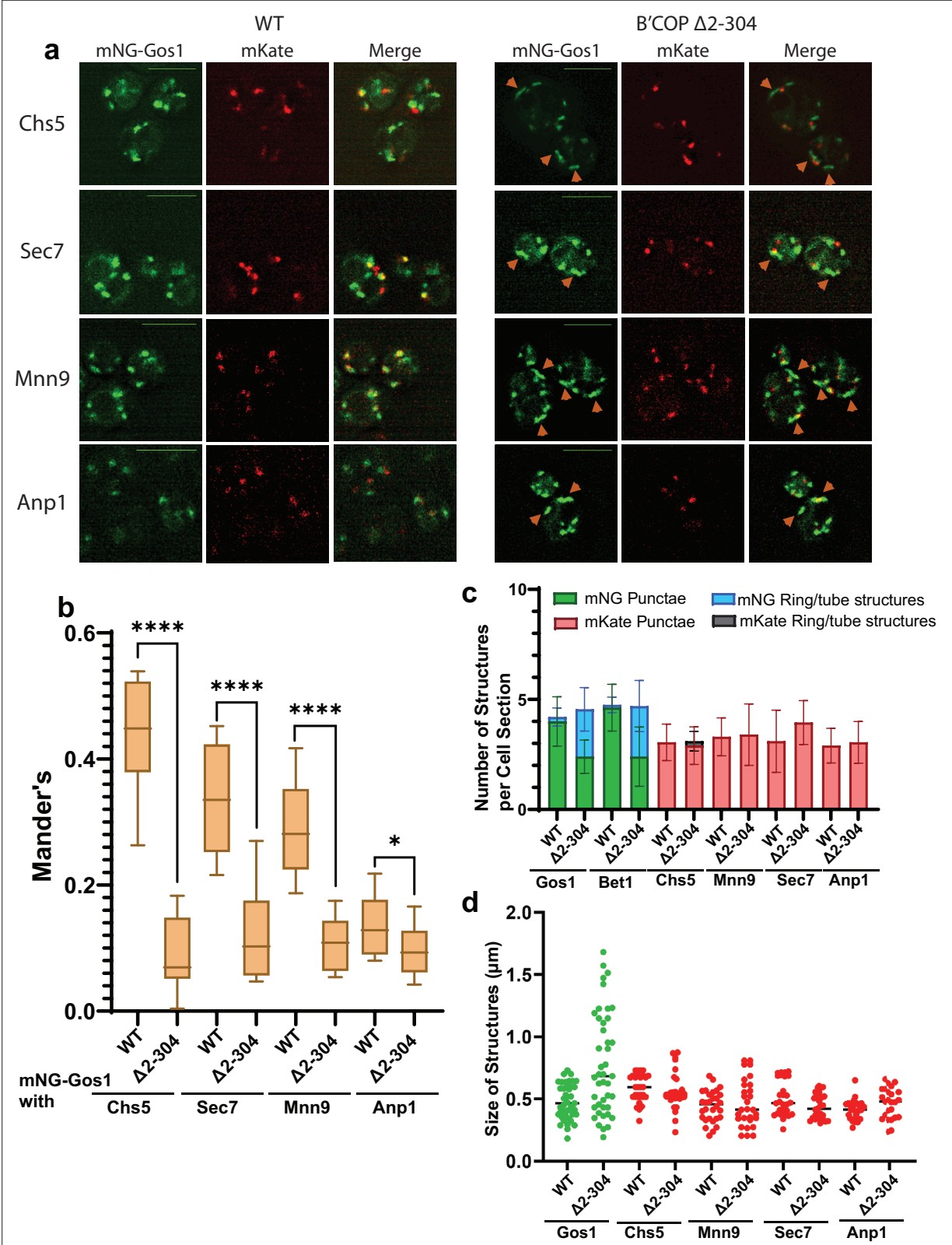

**Figure 2.** mNG-Gos1 accumulates in tube-like sequestration compartments lacking Golgi markers in β'-COP Δ2–304 cells. (**a, b**) Colocalization analysis of mNG-Gos1 with two late Golgi markers Chs5-mKate, Sec7-mKate and two early Golgi markers, Mnn9-mKate and Anp1-mKate indicates a general loss of mNG-Gos1 from early and late Golgi in β'-COP Δ2–304 cells compared to wild-type (WT) cells. Gos1 accumulates in tube-like structures (orange arrowheads) in the COPI mutant. Scale bars represent 5μm. Statistical analysis was done on means using three biological replicates, *t*-test

*Figure 2 continued on next page*

*Figure 2 continued*

(****p ≤ 0.0001,*p < 0.05, Ns p > 0.05). (**c**) The abundance of mNG-tagged Gos1 and Bet1 tube- or ring-like structures is increased β'-COP Δ2–304 cells compared to WT cells, but the total number of fluorescent structures is unchanged. The number and appearance of mKate-tagged Chs5, Sec7, Mnn9, and Anp1 structures are not altered in β'-COP Δ2–304 cells compared to WT cells. (**d**) The average size and size distribution of intracellular structures for mNG-Gos1 are significantly increased in β'-COP Δ2–304 cells compared to WT control but the average size and size distribution of mKate-tagged Chs5, Sec7, Mnn9, and Anp1 structures are not altered in β'-COP Δ2–304 cells compared to WT control.

The online version of this article includes the following figure supplement(s) for figure 2:

**Figure supplement 1.** Colocalization analysis of mNG-Bet1 indicates loss of Bet1 from early and late Golgi compartments in β'-COP Δ2–304 cells.

**Figure supplement 2.** Bet1 localizes to medial Golgi and does not transit the plasma membrane or ER.

**Figure supplement 3.** *cis*, medial, and *trans*-Golgi network morphology is normal in β'-COP Δ2–304 cells and Bet1 and Gos1 localize normally in β'-COP RKR cells.

**Figure supplement 4.** Aberrant structures to which mNG-Gos1 and -Bet1 accumulate in COPI mutant are not predominantly endosomal structures.

different for the Golgi markers in WT or β'-COP Δ2–304 cells (*Figure 2d*). For mNG-Gos1, the average length of the structures was significantly increased in β'-COP Δ2–304. Thus, the COPI mutation used here does not cause whole organelle-level changes in Golgi morphology or the localization of markers used here (*Figure 2*, *Figure 2—figure supplements 1 and 3*). Bet1 and Gos1 are substantially lost from Golgi compartments and mislocalized to tube- or ring-like structures (*Figures 1 and 2*, *Figure 2—figure supplements 1 and 2*).

To test if Bet1 and Gos1 were mislocalized to the endosomal system, we stained the WT and COPI mutant cells with FM-64 for 3 min to mark endosomal compartments. The majority of FM4-64-stained endosomes had a normal morphology in the mutant cells (*Figure 2—figure supplement 4*). We did see a subset of the mNG-SNARE tubular structures stain weakly with FM4-64. However, the degree of colocalization between the SNAREs and FM4-64 as measured by the Mander's colocalization coefficient was not significantly different between WT and COPI mutant cells. Therefore, it does not appear that Bet1 and Gos1 are being mislocalized from the Golgi to the endosomal system in the β'-COP Δ2–304 cells. We suspect that a failure to retrieve these SNAREs during maturation of the Golgi cisternae, combined with a normal retrieval of most resident Golgi proteins, results in Gos1 and Bet1 accumulating in dead-end compartments that we describe here as sequestration compartments.

The β'-COP-RKR mutant is incapable of binding dilysine motif cargoes through β'-COP. To determine whether Bet1 and Gos1 localization are dependent on the ability of β'-COP to bind dilysine cargoes, we compared the appearance and localization of mNG-Gos1 and Bet1 in WT or in β'-COP-RKR cells. No observable difference in the appearance or localization of Bet1 and Gos1 was observed. These observations suggest that the ability of β'-COP to bind ubiquitin but not the dilysine motifs is critical for proper localization of SNAREs Bet1 and Gos1 (*Figure 2—figure supplement 3b*).

## β'-COP binding to ubiquitin is essential for proper SNARE localization

The N-terminal WDR domain of β'-COP binds ubiquitin, and the COPI–ubiquitin interaction is critical for Snc1 retrieval (*Xu et al., 2017*). To determine whether mislocalization of other SNAREs in β'-COP Δ2–304 is due to the inability of β'-COP to bind ubiquitin, we used a set of COPI constructs (*Figure 3—figure supplement 1b–e*) where the N-terminal WDR domain of β'-COP was replaced with (1) a general UBD of Doa1 ($UBD_{Doa1}$), which is known to bind ubiquitin irrespective of the ubiquitin linkage type (*Mullally et al., 2006*) and (2) a UBD from Tab2 ($NZF_{Tab2}$) which specifically binds K63–ubiquitin linkages (*Moritsugu et al., 2018*). Compared to WT cells, Bet1, Gos1, and Snc1 were mislocalized to elongated tube structures in β'-COP Δ2–304 cells as seen previously (*Figure 1b*; *Figure 3a–f*). The replacement of the β'-COP 2–304 domain with the general UBD, $UBD_{Doa1}$ restored SNARE localization to punctate structures comparable to WT cells (*Figure 3a–f*). β'-COP-$UBD_{Doa1}$ does not restore the dilysine interaction (*Xu et al., 2017*); therefore, it is the ability of β'-COP to bind Ub that is critical for normal SNARE localization. Surprisingly, however, the K63-linkage restricted β'-COP $NZF_{Tab2}$ construct did not significantly correct the Bet1 or Gos1 localization pattern. We previously found that the β'-COP Δ2–304 Snc1 recycling defect was corrected by replacing the WDR domain with either the $UBD_{Doa1}$ or $NZF_{Tab2}$ (*Xu et al., 2017*). Consistently, we found here that both the $UBD_{Doa1}$ and $NZF_{Tab2}$ constructs significantly restored the WT pattern of intracellular structures labeled with mNG-Snc1 (*Figure 3b, e*). However, even though Snc2 is functionally and evolutionarily closely related

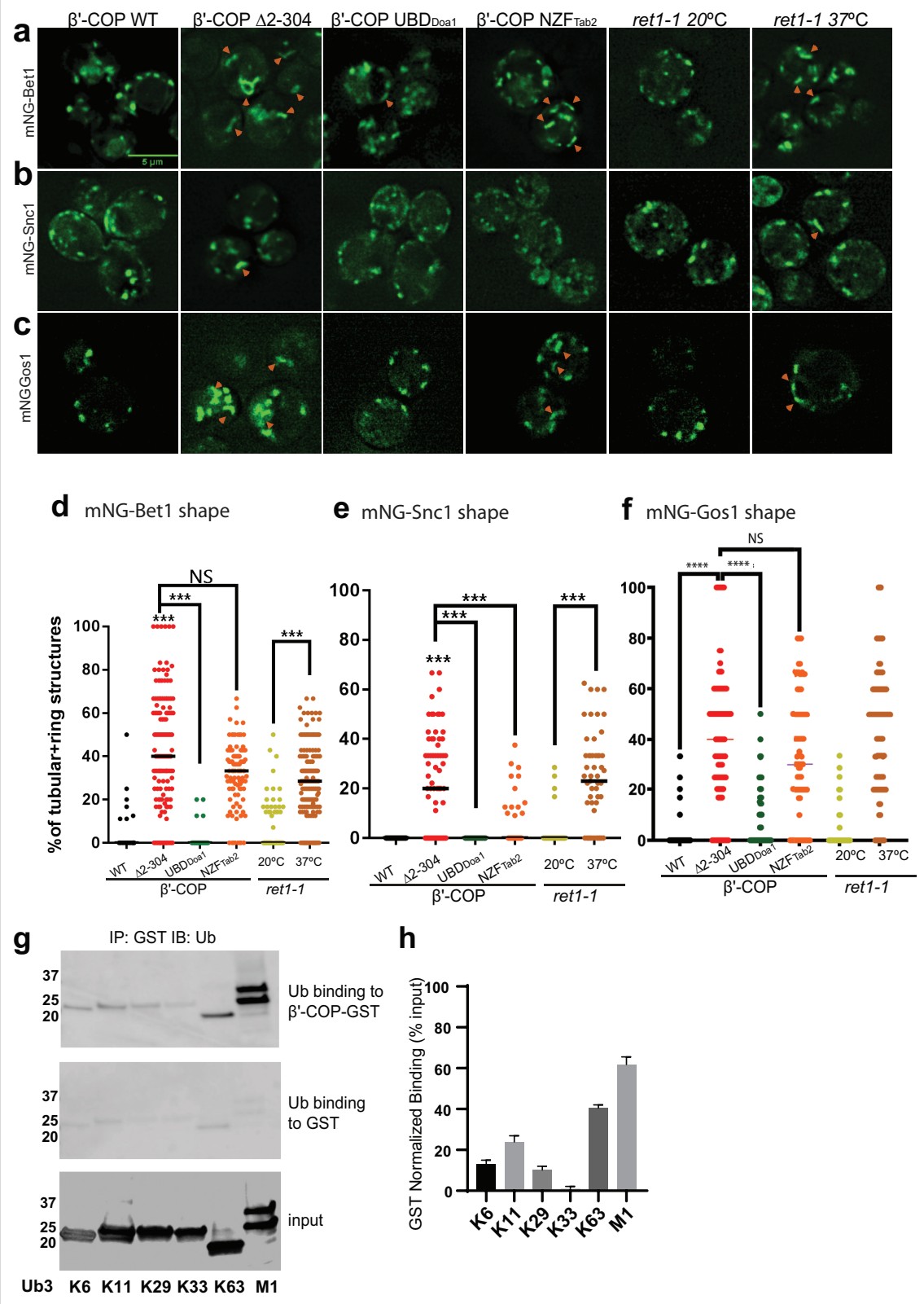

**Figure 3.** β'-COP binding to ubiquitin is critical for proper SNARE localization. Deletion of the N-terminal ubiquitin-binding WDR of β'-COP (Δ2–304) leads to mislocalization of (**a**) mNG-Bet1, (**b**) mNG-Snc1, and (**c**) mNG-Gos1 into elongated tubular and ring-like structures (orange arrowheads). This phenotype is rescued by the replacement of the N-terminal ubiquitin-binding WDR of β'-COP by the general ubiquitin-binding domain Doa1 (β'-COP UBD_{Doa1}) (*Figure 3—figure supplement 1*). The replacement of N-terminal UBD of β'-COP with K63-specific UBD, NZF_{Tab2} (β'-COP NZF_{Tab2}) rescues the

*Figure 3 continued on next page*

*Figure 3 continued*

mislocalization phenotype for Snc1 but not for Bet1, Gos1, and Snc2 (*Figure 3—figure supplement 2*). The mislocalization of Bet1 and Snc1 observed in β'-COP Δ2–304 cells comparable to COPI inactivation phenotype observed for *ret1-1* at nonpermissive temperatures. (**d–f**) Statistical differences were determined using a one-way analysis of variance (ANOVA) on the means of the three biological replicates (****p<0.0001, ***p < 0.001). (**g**) GST-β'-COP (1–604) binds linear and K63-linked triUb and to some extent to K6, K11, and K29 triUb relative to the GST-only control. 0.5 mM of GST and GST-tagged WDR proteins immobilized glutathione beads were incubated 250 nM Ub3 for corresponding linkages. (**h**) Quantitation of Ub3 polymers binding (GST-only background subtracted) relative to input. The values represent mean ± standard error of the mean (SEM) from three independent binding experiments. Scale bar represents 5 μm.

The online version of this article includes the following figure supplement(s) for figure 3:

**Figure supplement 1.** COPI model figure with β'-COP constructs that differ in their ability to recognize and bind ubiquitin linkages.

**Figure supplement 2.** The ability of β'-COP to bind ubiquitin is critical for the proper localization of Snc2.

to Snc1, we found that $UBD_{Doa1}$ restored the mNG-Snc2 WT pattern, but the K63-restricted $NZF_{Tab2}$ domain did not (*Figure 3—figure supplement 2*). For GFP-Sec22, β'-COP-$UBD_{Doa1}$ fully prevented vacuolar mislocalization while a partial rescue was conferred by β'-COP $NZF_{Tab2}$ (*Figure 1—figure supplement 1c, d*). Thus, Snc1 and Sec22 can use K63-linked polyUb chains for their trafficking, but Bet1, Gos1, and Snc2 appear to rely on COPI binding to some other ubiquitin linkage type.

Next, we examined the localization of mNG-tagged Bet1, Snc1, and Snc2 in a temperature-sensitive COPI mutant (*ret1-1*) grown at the permissive temperature and shifted to the nonpermissive temperature of 37°C for 1 hr. The *ret-1* mutation is within α-COP and substantially inactivates all known COPI functions (*Gaynor et al., 1998*; *Ishii et al., 2016*; *Letourneur et al., 1994*). Bet1, Snc1, and Snc2 were observed in tubular and ring-like structures in *ret1-1* at the nonpermissive temperature (*Figure 3a–f*, *Figure 3—figure supplement 2*). Interestingly, the localization pattern seen for Bet1, Snc1, and Snc2 in β'-COP Δ2–304 cells was comparable to *ret1-1* at the nonpermissive temperature (*Figure 1b–c*; *Figure 3a–c*). These data indicate that perturbations in the ability of β'-COP to bind ubiquitin in β'-COP Δ2–304 substantially disrupt COPI function with respect to Bet1, Snc1, and Snc2 localization.

β'-COP has been shown to bind K63-polyUb chains but not K48-polyUb or monoubiquitin (monoUb) (*Xu et al., 2017*). Since a general UBD rescued the localization for all 4 SNAREs, but not K63-specific UBD (*Figure 3a–f*), we reasoned that β'-COP might be able to bind other polyUb chains. To test this hypothesis, we assayed the ability of heterologously purified GST-tagged β'-COP to bind K6-, K11-, K29-, K33-, and linear (M1)-linked polyUb chains. K63-polyUb was used as a positive control, and GST-only was used to determine background levels of ubiquitin binding to GST (*Figure 3g, h*). β'-COP is capable of binding linear ubiquitin chains and more weakly to K6-, K11-, and K29-polyUb chains (*Figure 3g, h*).

## Fusion of a deubiquitinase domain to COPI leads to SNARE mislocalization

To analyze the functional significance of ubiquitination within the COPI–SNARE system, we designed constructs where a deubiquitinase domain, UL36 (DUB) from Herpes Simplex Virus 1 (*Kattenhorn et al., 2005*), was fused to either α- or β'-COP. A catalytically dead version of UL36 (DUB*) wherein an active site Cys is mutated to Ser and thus cannot deubiquitinate substrates was engineered as a control. Strains expressing COPI-DUB constructs, irrespective of whether α- or β'-COP was fused to DUB, were enlarged in size (*Figure 4a, b*). Additionally, we observed mislocalization of Bet1 and Gos1 in COPI-DUB constructs wherein mNG-tagged SNAREs were observed in enlarged punctate structures, elongated tube structures, or ring-like structures (*Figure 4a, c*). COPI-DUB* constructs did not display significant phenotypic changes. The fusion of a DUB domain to COPI phenocopies the mislocalization pattern for Bet1 and Gos1 in the COPI (*ret1-1*) mutant at nonpermissive temperatures, supporting the importance of ubiquitination in COPI-mediated regulation of SNARE localization.

## Ubiquitination is associated with Gos1, Ykt6, and Sed5 SNARE complexes

Global analyses of the budding yeast proteome have identified ubiquitinated lysines in Gos1, Snc1, and Snc2 but not Bet1 (*Swaney et al., 2013*). We set out to test if ubiquitination could be detected by immunoprecipitating the SNAREs and probing for ubiquitin on immunoprecipitated samples and

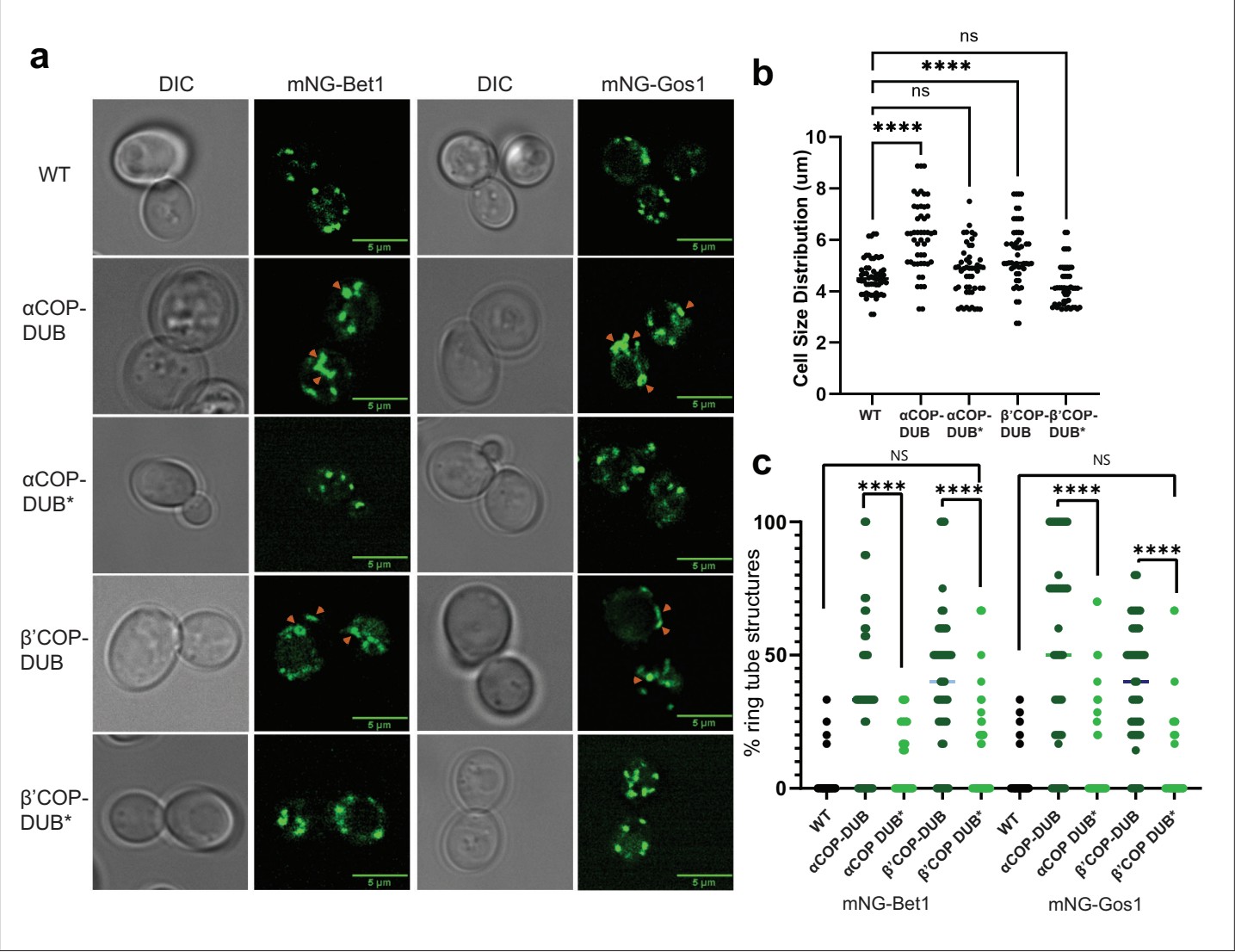

**Figure 4.** Deubiquitinase fusion to COPI subunits causes SNARE mislocalization. mNG-tagged Bet1 and Gos1 were imaged in cells in which a deubiquitinase domain (DUB) was fused to the C-terminus of α- and β'-COP to generate αCOP-DUB and β'-COP-DUB, respectively, along with catalytically dead controls αCOP-DUB* and β'-COP-DUB*. (**a, b**) Cells carrying COPI-DUB fusion were larger in size compared to wild-type (WT) cells as well as catalytically dead controls . (**a, c**) Significant accumulation of Bet1 and Gos1 in the elongated tube- or ring-like or enlarged punctate structures (orange arrows) was observed in αCOP-DUB and β'-COP-DUB backgrounds compared to corresponding DUB* control or WT cells . Statistical differences were determined using a one-way analysis of variance (ANOVA) on the means of the three biological replicates (****p < 0.0001).

by detecting the pooled ubiquitin released off of immunoprecipitated samples following a deubiquitinase (DUB) treatment. We individually tagged Bet1, Gos1, and Snc1 with 6xHIS-TEV-3xFLAG at their C-termini by chromosomal integration of the tag constructs. Following FLAG immunoprecipitation, the samples were treated with mock buffer (no DUB) or deubiquitinases (DUB) (*Figure 5a*) and probed with FLAG (*Figure 5b*) or ubiquitin antibodies (*Figure 5c*). Art1, a ubiquitinated protein from *S. cerevisiae*, was used as a positive control, and untagged cells (Ctrl) were used as a negative control. The FLAG antibody recognizes a nonspecific band at approximately 20 kDa (*Figure 5b*, Ctrl Lane) that unfortunately comigrates with Bet1-FLAG and Snc1-FLAG as indicated by the increased band intensity at 20 kDa in those samples relative to the untagged control (Ctrl) sample. In addition, Bet1-FLAG exhibited a significant smear extending to greater than 40 kDa (*Figure 5b*). However, this smeared pattern for Bet1-FLAG was not collapsed by DUB treatment, nor was this smear recognized by the antiubiquitin antibody. Moreover, the amount of monoUb released from Bet1-FLAG by DUB treatment was not significantly different from the control sample (*Figure 5c, d*).

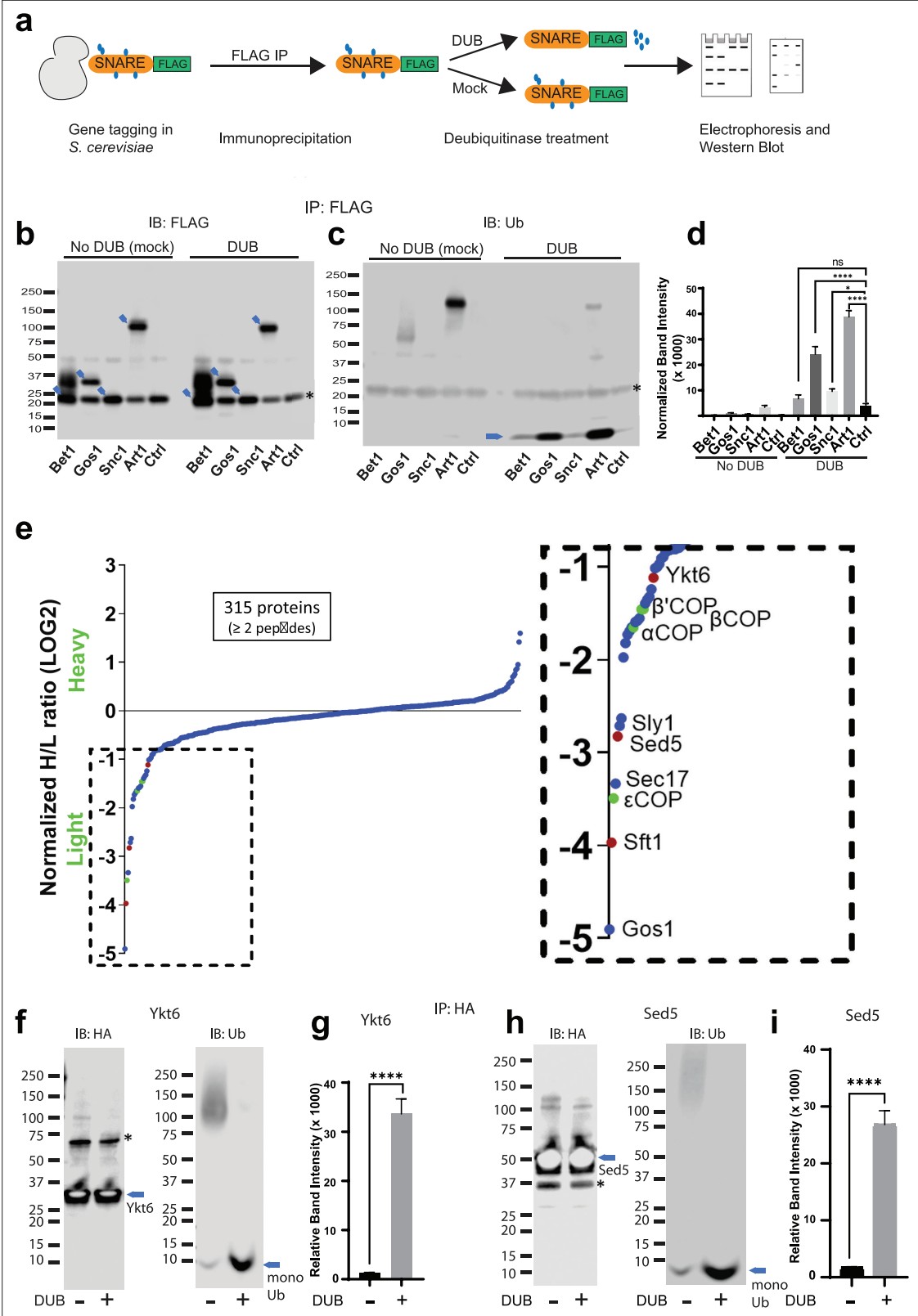

**Figure 5.** Multiple Golgi SNARE complexes are modified with ubiquitin. (**a**) Schematic of the experimental setup wherein SNAREs were individually tagged with FLAG and immunoprecipitated using anti-FLAG beads. Half the samples were mock treated, and the other half was treated with deubiquitinase (DUB). Western blots of samples are probed with FLAG (**b**) or ubiquitin antibody (**c**). Blue arrows in (**b**) indicate the position of FLAG-tagged protein and the asterisk indicates the position of a background band. (**d**) Quantitation of the amount of monoubiquitin released from the

*Figure 5 continued on next page*

*Figure 5 continued*

samples by deubiquitinases. (**e**) SILAC mass spectrometric analysis of Gos1-FLAG pulldown samples indicates enrichment of SNAREs Sft1, Ykt6, and Sed5 (red dots) and COPI subunits (green dots) with Gos1 . (**f–i**) Western blot analysis showing HA-tagged Ykt6, and Sed5 probed for ubiquitination following HA immunoprecipitation and deubiquitinase treatment. (**g, i**) Quantitation of monoubiquitin. Statistical differences were determined using a one-way analysis of variance (ANOVA) with multiple comparison test on three biological replicates (****p ≤ 0.0001, *p < 0.05, Ns p > 0.05).

The online version of this article includes the following figure supplement(s) for figure 5:

**Figure supplement 1.** Ubiquitination associated with COPI, Gos1, and Glo3 is non-K48 linked.

**Figure supplement 2.** Ubiquitin associated with Gos1, α-COP, Ykt6, and Sed5 is not K63- or M1-linked polyubiquitin.

For Gos1-FLAG immunoprecipitations probed with antiubiquitin antibody, a smeared pattern was observed in the 50–80 kDa molecular weight region (*Figure 5b*) when probed with anti-Ub, which collapsed, releasing a significant amount of monoUb following DUB treatment (*Figure 5c, d*, Gos1 lanes). A similar smeared pattern is seen for Art1 in mock-treated samples around 75–130 kDa molecular weight region, which was converted to monoUb by DUB treatment (*Figure 5c, d*, Art1 lanes). Although the smeared pattern for Snc1 was not apparent in these samples, DUB treatment released more monoUb than control samples (*Figure 5d*, Snc1). We initially focused our attention on Gos1 because it appeared to be ubiquitinated and evidence for the importance of Snc1 ubiquitination has already been reported (*Chen et al., 2011*; *Xu et al., 2017*; *Figure 5c, d*).

To identify other proteins specifically associated with Gos1 when purified under conditions that preserved ubiquitination, we employed a Stable Isotope Labeling by/with Amino acids in Cell culture (SILAC) mass spectrometry (MS) approach. A strain expressing Gos1-FLAG was grown in a light isotope medium and untagged control cells used to determine the nonspecific background proteins in the FLAG IP, were grown in a heavy isotope medium. Importantly, the samples were processed in the presence of DUB inhibitors to preserve ubiquitination on Gos1 and other proteins in the samples. Gos1 is reported to form a functional t-SNARE complex with Ykt6 and Sed5 that mediates fusion with intra-Golgi retrograde vesicles bearing Sft1 (*Parlati et al., 2002*). We observed significant enrichment of peptides from these partner SNAREs with Gos1-FLAG and known SNARE regulators like Sec17 and Sly1 (*Demircioglu et al., 2014*; *Song et al., 2021*; *Figure 5e*). Importantly, we also found several COPI subunit peptides that were enriched to comparable levels as Ykt6, Sft1, and Sed5 in the Gos1 pulldown samples (*Figure 5e*).

To probe the ubiquitination status of Gos1-binding SNARE partners, we individually tagged Ykt6 and Sed5 with 3xHA tag on the N-terminus (attempts at C-terminally tagging Ykt6 and Sed5 were unsuccessful potentially owing to structurally/functionally important modifications at the C-terminus, such as Ykt6 palmitoylation). HA-tagged Ykt6 and Sed5 were immunoprecipitated using anti-HA and probed for their ubiquitination status. A smeared pattern associated with ubiquitination was observed for both Ykt6 (*Figure 5f*) and Sed5 (*Figure 5h*) in mock-treated samples, which was collapsed by DUB treatment to monoUb (*Figure 5f–i*). These data support previously published high-throughput results indicating that Gos1, Ykt6, and Sed5 are ubiquitinated (*Swaney et al., 2013*). The differences in the size distributions of polyUb smear in each SNARE immunoprecipitate suggest that this assay is primarily detecting direct modification of Gos1, Ykt6, and Sed5 as opposed to the aggregate polyUb associated with the entire SNARE complex.

## Nondegradative ubiquitination is associated with Gos1, COPI, and Glo3 complexes

We observed significant enrichment of COPI subunits in the Gos1 pulldown samples analyzed with SILAC MS (*Figure 5e*). Therefore, we probed the ubiquitination status of FLAG-tagged COPI (α- and β'-COP subunits) and Glo3, as this ArfGAP is reported to bind COPI and SNAREs (*Rein et al., 2002*). The FLAG IPs probed with antiubiquitin antibody show a substantial amount of monoUb released from COPI and Glo3 immunoprecipitates following the DUB treatment (*Figure 5—figure supplement 1a–e*). K48-linked polyUb chains are known to target proteins for proteasomal degradation. To address whether Gos1, COPI, and Glo3 complexes are modified with K48-linked polyUb, we treated the samples with a K48-specific DUB. No significant change in the smeared electrophoretic pattern or the release of monoUb in the samples was observed with or without K48-specific DUB

treatment suggesting that the ubiquitination associated with COPI and Glo3 is not a degradation signal (*Figure 5—figure supplement 1a-e*).

We also probed Gos1, Ykt6, Sed5, COPI, and Glo3 FLAG-immunoprecipitated samples with K63-specific deubiquitinase (K63-DUB [*Sato et al., 2008*]), linear ubiquitin-specific deubiquitinase (M1-DUB [*Keusekotten et al., 2013*]), or a general deubiquitinase (DUB [*Baek et al., 1997*]) as a control (*Figure 5—figure supplement 2*). No significant release of ubiquitin was observed following K63- or M1-DUB treatment compared to the untagged control (*Figure 5—figure supplement 2*). A detectable amount of ubiquitin was released from Gos1 following K63-DUB treatment, but the signals were not significantly above the background levels (*Figure 5—figure supplement 2*). A significant level of released ubiquitin was detected for these samples when treated with the general deubiquitinase. The lack of K63 linkages on these components is also consistent with live-cell imaging data (*Figure 3a–f*), showing that β′-COP with a K63-specific binding domain failed to support the trafficking of Bet1, Gos1, and Snc2. Thus, the ubiquitination associated with Gos1, COPI, and Glo3 complexes appears to be nondegradative (non-K48 or non-K63) in nature and may modulate protein interactions in the COPI-dependent retrieval of SNAREs within the Golgi.

## Ubiquitination stabilizes Golgi SNARE–COPI complexes

To explore the possibility that ubiquitination is an important regulator of protein–protein interactions in the COPI–SNARE system, we used comparative pulldown studies using FLAG-tagged SNAREs under conditions that preserved endogenous ubiquitination (w Ub) or catalyzed removal of ubiquitin (w/o Ub) (*Figure 6a*). An equal amount of Gos1-FLAG was pulled down in both w Ub and w/o-Ub conditions (*Figure 6c*). Probing samples with a ubiquitin antibody showed a ubiquitin smear associated with Gos1 immunoprecipitated using 'w Ub' conditions, most of which was stripped off under 'w/o--Ub' conditions (*Figure 6b*). We next probed these samples with COPI and Arf antibodies. Significant enrichment of COPI subunits and Arf was observed with Gos1 when ubiquitination was preserved, compared to 'w/o-ub' conditions (*Figure 6e–g*). The ubiquitination, thus, appears to play a role in the assembly and/or stability of COPI coatomer complex with Gos1.

Our assays and previous reports indicate that Gos1 is ubiquitinated (*Swaney et al., 2013*), but ubiquitination has not been detected on Bet1 (*Figure 5b–d*). Loss of the UBD of COPI in β′-COP Δ2–304 led to mislocalization of both Bet1 and Gos1 to the elongated tube- and ring-like structures (*Figures 1b and 3a, c*). Therefore, we tested whether ubiquitination affects the interaction of Bet1 with COPI coat complex components. The smeared pattern associated with Bet1 in the blot probed with anti-FLAG antibody is similar under 'w Ub' and w/o Ub' conditions (*Figure 6—figure supplement 1b,c*). Nonetheless, we see the enrichment of COPI subunits with Bet1 under 'w Ub' conditions compared to 'w/o Ub' (*Figure 6—figure supplement 1d*). Similarly, Arf is significantly enriched with Bet1 when ubiquitin was present on the complexes (*Figure 6—figure supplement 1e, f*). Control experiments indicated that the presence of DUB inhibitors during cell lysis was most critical to preserve the SNARE–COPI complex (*Figure 6a*, *Figure 6—figure supplement 1g, h*). Therefore, the role of ubiquitination in the assembly and stability of COPI coatomer complex with Bet1 appears to function independently of the Bet1 ubiquitination status. Altogether, the data reveal ubiquitin-mediated stabilization of COPI–Golgi SNARE complexes.

## Glo3 is not enriched in ubiquitin-stabilized SNARE–COPI–Arf complexes and Gos1 localization is unaffected in *glo3Δ* cells

Glo3 is proposed to be part of a SNARE–Arf–COPI priming complex, but we failed to detect any Glo3 peptides in the Gos1 immunoprecipitates by MS (*Figure 5e*). To further test whether ArfGAP Glo3 is present in the ubiquitin-stabilized SNARE–COPI–Arf complex, we performed Gos1-FLAG pulldowns under w Ub and w/o-Ub conditions in cells expressing Glo3 C-terminally tagged with GST. We detected a small amount of Glo3-GST coimmunoprecipitating with FLAG-Gos1, but no significant difference was observed in the presence or absence of Ub (*Figure 7a, c*). In contrast, association of Arf with Gos1 was significantly enriched using w Ub conditions compared to w/o-Ub conditions (*Figure 7a, d*). Cell lysate controls probed for GST in cells expressing only Gos1-FLAG or both FLAG-Gos1 and Glo3-GST confirmed the identity of the Glo3-GST band (*Figure 7b*).

To determine whether we can detect Glo3 interaction with COPI, we performed SILAC coimmunoprecipitation analysis using C-terminally FLAG-tagged α-COP under ubiquitin preserved conditions.

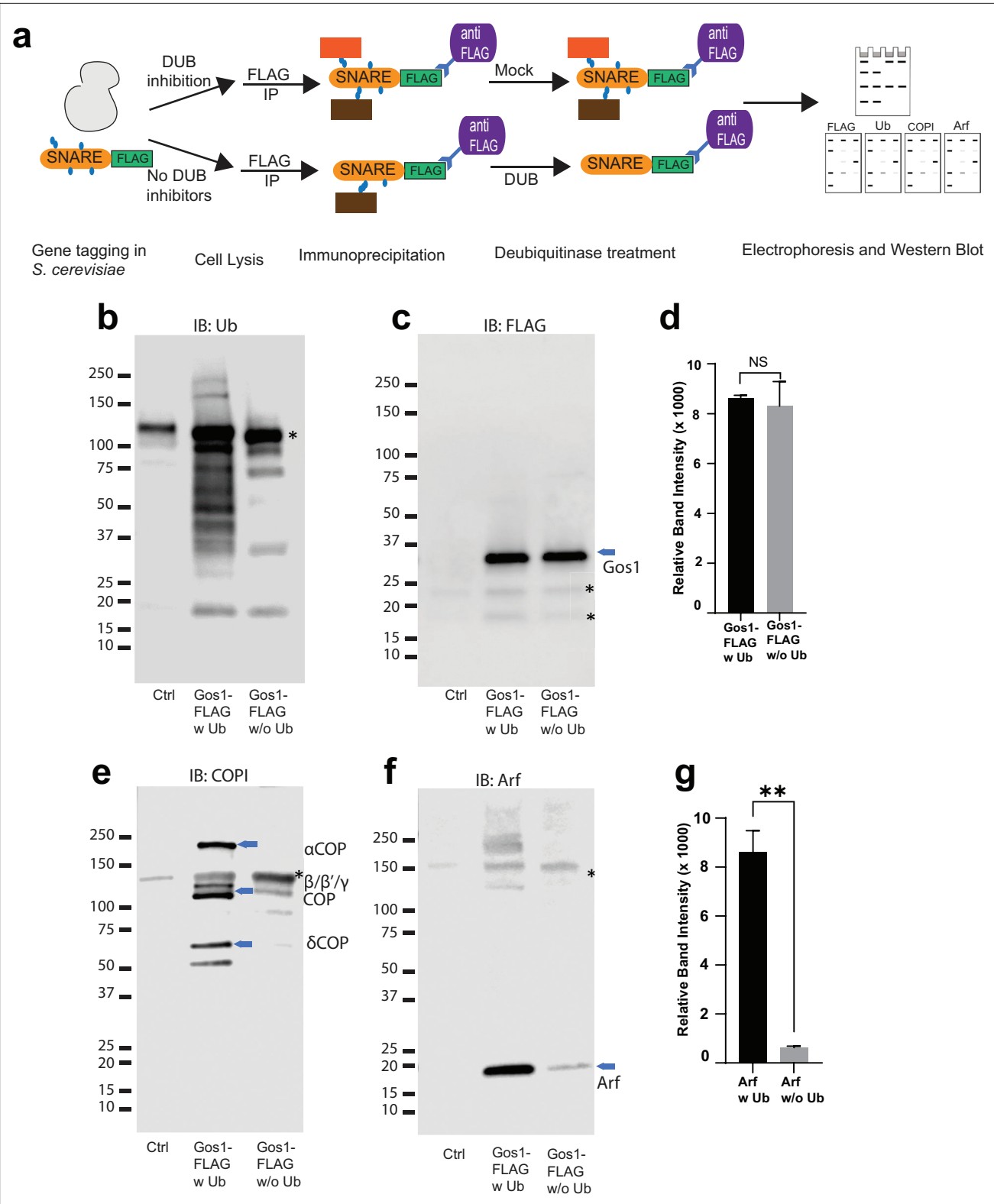

**Figure 6.** Ubiquitin modification stabilizes a priming complex between COPI, Arf, and SNAREs. (**a**) Schematic of the experimental setup wherein FLAG-tagged SNAREs are divided into two equal portions, and one is processed under 'w Ub' conditions (DUB inhibitors used during the lysis step and no deubiquitinases [mock] treatment) and the other portion is processed under 'w/o Ub' condition (no deubiquitinase inhibitors used during lysis and immunoprecipitated samples are treated with deubiquitinases). (**b–g**) Western blot data showing comparative pulldowns of Gos1-FLAG (**b–f**) and Bet1-FLAG (*Figure 6—figure supplement 1*) processed under 'ubiquitin-preserved' (w Ub) and 'no-ubiquitin' (w/o Ub) condition, and probed for Ub

*Figure 6 continued*

(**b**), FLAG (**c**), COPI (**e**), and Arf (**f**). Untagged cells processed under 'w UB' condition to determine background binding were used as a control (Cntr) and abundant background bands are marked with an asterisk. Quantitation of Gos1-FLAG (**d**) and Arf (**g**) in the pulldown samples. Band intensities are measured using ImageJ. Quantitation was done on three biological replicates using a *t*-test (**p < 0.01, Ns p > 0.05).

The online version of this article includes the following figure supplement(s) for figure 6:

**Figure supplement 1.** Ubiquitin-mediated enhancement of COPI and Arf Bet1.

As expected, Glo3 was enriched with COPI, indicating eventual recruitment of Glo3 onto the COPI coat (*Figure 7f*). The COPI sample was also enriched for several ER residence membrane proteins bearing C-terminal dilysine motifs (*Figure 7f*, red datapoints). However, no SNAREs were coimmunoprecipitated with COPI, indicating that the COPI complexes recovered here were significantly different from SNARE–COPI complexes that contained Gos1.

In addition, we analyzed mNG-Gos1 localization in *glo3Δ* cells with WT and β'-COP Δ2–304 cells as control. Compared to WT cells, no significant changes in the size or distribution of mNG-Gos1 punctae were observed in *glo3Δ* (*Figure 7g*). As reported earlier, Gos1 is mislocalized to elongated tube-like structures in β'-COP Δ2–304 cells. Together, these data suggest that Glo3 can weakly bind Gos1 but unlike COPI, this interaction is not stabilized by the presence of Ub. In addition, Glo3 does not appear to be required for COPI-dependent Gos1 localization.

Altogether, these results indicate that ubiquitin plays critical role in stabilizing a complex between SNAREs, COPI, and Arf that is important for COPI function in retrieving a subset of SNAREs. Glo3 appears to be present at low levels in the Gos1–Arf–COPI complex, but this interaction is not modulated by ubiquitination.

## Discussion

We previously discovered that COPI binds specifically to polyUb chains, and that this interaction is crucial for recycling Snc1, an exocytic v-SNARE, back to the TGN. In this study, we broadly probed the role of COPI–ubiquitin interactions on the localization of 16 additional budding yeast SNAREs to determine whether ubiquitination of coat components is a general mechanism for SNARE sorting. While localization of most mNG-tagged SNAREs was unaffected by deletion of the ubiquitin-binding N-terminal WDR domain of β'-COP, we found a significant change in the localization pattern for Bet1, Gos1, Snc1, Snc2, and partially for Bos1, Tlg1, and Sec22 in β'-COP Δ2–304 (*Figures 1b, c and 3a–f*, *Figure 1—figure supplement 1*). Normal SNARE localization is restored by the replacement of β'-COP N-terminal WDR domain (β'-COP-UBD) with an unrelated UBD (*Figure 3*, *Figure 3—figure supplement 2*). Moreover, we found nondegradative (non-K48 and -K63) ubiquitin associated with multiple Golgi SNAREs (Gos1, Ykt6, and Sed5) and the COPI machinery (*Figure 5b–d, f–i*, *Figure 5—figure supplements 1 and 2*), and that these ubiquitin modifications were essential for the stabilization of COPI, Arf, and Golgi SNARE complexes (*Figure 6*). For Gos1, the ubiquitin-stabilized SNARE–COPI–Arf complex is not enriched for the ArfGAP Glo3 (*Figure 7*). These studies highlight the important role of ubiquitination in COPI-mediated trafficking, specifically in the regulation of Golgi SNARE localization.

SNAREs are essential components of the endomembrane system where they play a major role in determining the directionality and specificity of vesicular transport. Our data provide exciting insights into regulation of SNARE localization required to maintain a functional endomembrane system. In the case of Sec22, the failure of COPI-mediated retrieval of this SNARE causes mislocalization to the vacuole. Bet1 and Gos1, however, display an unusual mislocalization phenotype from normal punctate Golgi in WT cells to abnormal tubular and ring-shaped structures that lack Golgi markers in β'-COP Δ2–304 cells. Although the degree of colocalization with Golgi markers is substantially reduced in this COPI mutant, neither Bet1 or Gos1 significantly accumulate at downstream compartments such as the plasma membrane, endosomes, or vacuoles. We propose that Bet1 and Gos1 accumulate in dead-end sequestration structures in β'-COP Δ2–304 that are remnants of the Golgi cisternal maturation process. None of the Golgi marker proteins assessed (Sed5, Anp1, Mnn9, Rer1, Aur1, Chs5, or Sec7) localized to the same abnormal structures containing Bet1 or Gos1 or showed any evidence of being mislocalized (this study and *Xu et al., 2017*). These data indicate that COPI is still functioning well to recycle most proteins from late Golgi compartments back to early compartments or the ER in

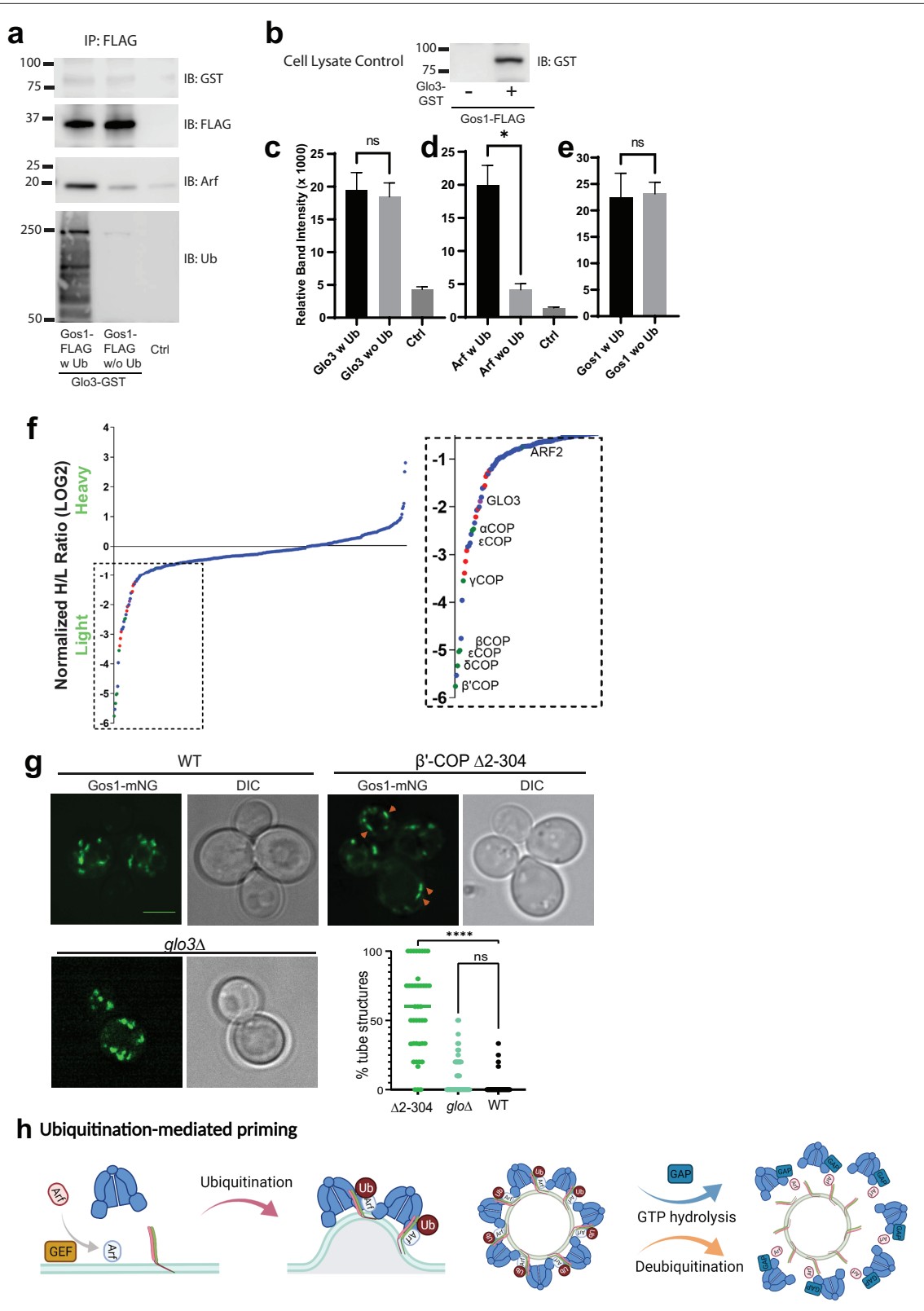

**Figure 7.** ArfGAP is not enriched in ubiquitin-stabilized SNARE–Coat complexes and is not required for Gos1 localization. (**a–e**) Western blot data showing comparative pulldowns of Gos1-FLAG from cells expressing FLAG-tagged Gos1 and GST-tagged Glo3. Samples were processed under 'ubiquitin-preserved' (w Ub) and 'no-ubiquitin' (w/o Ub) condition, and probed for Glo3 (anti-GST), Gos1 (anti-FLAG), Arf, and Ub. Untagged cells processed under 'w UB' condition to determine background binding were used as a control (Ctrl). Cell lysates from cells expressing only FLAG-tagged

*Figure 7 continued on next page*

*Figure 7 continued*

Gos1 or both FLAG-tagged Gos1 and GST-tagged Glo3 probed with anti-GST antibody are included as controls to ensure expression of GST-tagged Glo3. Quantitation of (**c**) Glo3-GST, (**d**) Arf, and (**e**) Gos1-FLAG samples. Band intensities are measured using ImageJ. Quantitation was done on three biological replicates using a *t*-test (*p < 0.05, Ns p > 0.05). (**f**) SILAC mass spectrometric analysis of αCOP-FLAG pulldown samples indicate the enrichment of other COPI subunits (green dots), ArfGAP Glo3 (purple dots), and dilysine COPI cargo (red dots) but no SNAREs . (**g**) Live cell imaging of mNG-Gos1 in wild-type (WT), β'-COP Δ2–304 and *glo3Δ* cells. Quantification of % tube structures for each strain was from three biological replicates with 60 or more cells analyzed for each sample . Statistical differences were determined using a one-way analysis of variance (ANOVA) on the means of the three biological replicates (****p < 0.0001). Scale bar represents 5 μm. (**h**) Model showing ubiquitination-mediated priming of a SNARE–Arf–COPI complex. Glo3 is recruited at later stages after vesicle budding leading to hydrolysis of Arf-GTP and disassociation of COPI complex.

β'-COP Δ2–304 cells. A failure of COPI to retrieve Gos1 and Bet1, and perhaps an inability of these SNAREs to enter vesicles destined for the plasma membrane or endocytic system, could lead to the accumulation of compartments that house only a few mislocalized proteins. It is also possible that Gos1 and Bet1 are transported to endosomes in β'-COP Δ2–304 cells but are very efficiently retrieved by retromer or sorting nexins. In this case, the abnormal Gos1 or Bet1 structures could represent functional intermediates that have lost most of their Golgi identity and only weakly stain with the endocytic tracer FM4-64. The appearance of abnormal Gos1 and Bet1 structures is almost identical in a COPI temperature-sensitive mutant that displays a broader defect in protein trafficking. This suggests that the β'-COP Δ2–304 cells have substantially lost COPI function in sorting Bet1 and Gos1, which may underpin the slow growth phenotype of this mutant. In contrast, β'-COP-UBD$_{Doa1}$ and β'-COP-RKR cells sort Gos1 and Bet1 normally, even though these β'-COP variants cannot bind to dilysine cargoes, and both variants support WT growth of yeast. Thus, the Gos1 and Bet1 phenotypes observed in β'-COP Δ2–304 are directly caused by a deficiency in β'-COP binding to polyUb and are not a secondary effect of disrupting the interaction with dilysine motifs.

The type of ubiquitin linkages required for intra-Golgi SNARE interaction with COPI appears to be different from the ubiquitin linkages required to sort Snc1. β'-COP binds preferentially to K63-linked polyUb chains and does not bind monoUb, diUb, or K48-linked polyUb chains (*Xu et al., 2017*). The replacement of the N-terminal WDR domain of β'-COP with the NZF domain from Tab2, which binds specifically to K63-linked polyUb, substantially restores Snc1 trafficking to the plasma membrane (*Xu et al., 2017*; *Figure 3b*). Surprisingly, the β'-COP-Tab2$_{NZF}$ fusion fails to support the trafficking of Bet1 or Gos1 and only partially supports the trafficking of Sec22 (*Figure 3a–f*, *Figure 1—figure supplement 1*). We further explored the binding specificity of β'-COP and found that it is also capable of binding linear polyUb chains and to K6-, K11-, and K29- polyUb chains (*Figure 3g, h*). Moreover, the polyUb chains detected in the SNARE or COPI pulldowns are resistant to M1-, K63-, or K48-specific DUBs (*Figure 5—figure supplement 2*). Thus, the ubiquitin modifications present are unlikely to be targeting COPI to the proteasome or the SNAREs to the vacuole for degradation. Our data provide compelling evidence that the nonproteolytic ubiquitin code regulates the COPI-dependent trafficking patterns for Golgi SNAREs.

PolyUb chains on SNAREs could form a sorting signal that COPI recognizes in order to recycle them from downstream compartments, as previously proposed for Snc1 (*Xu et al., 2017*). However, several observations in the current study suggest broader roles of ubiquitination in regulating SNARE trafficking. For example, the Golgi localization of Bet1 relies on COPI's ability to bind ubiquitin, but we could not detect ubiquitinated forms of this SNARE (*Figure 5b–d*). However, ubiquitination was associated with other SNAREs, including Gos1, Ykt6, and Sed5, multiple COPI subunits, and ArfGAP Glo3 (*Figure 5b–d, f–i*, *Figure 5—figure supplements 1 and 2*). It is possible that Bet1 associates with another cargo protein that is ubiquitinated, and the ubiquitin serves as the COPI-dependent sorting signal for both proteins. It is also possible that ubiquitination induces conformational changes in COPI driven by β'-COP interaction with ubiquitin attached to itself or to other COPI subunits. Such a COPI conformational change could produce a high-affinity binding site for Bet1. The role of ubiquitin in mediating the stability of the COPI–SNARE complex is further supported by the observation that Bet1 and Gos1 are mislocalized when ubiquitin is stripped from COPI–SNARE system by fusing a deubiquitinase domain to COPI components (*Figure 4*).

We were surprised to find that COPI was co-enriched with Gos1-FLAG in the SILAC-based MS data (*Figure 5e*) because cargo–coat interactions are typically low affinity. Arf1 was also present in this dataset, although not as highly enriched as the COPI subunits. We considered the possibility that the

conditions used to pulldown Gos1-FLAG that preserve ubiquitination may have stabilized the COPI–Gos1 interaction. Indeed, performing these Gos1-FLAG pulldowns in the presence of active DUBs to remove ubiquitin dramatically reduces the amount of COPI and Arf recovered with Gos1-FLAG relative to samples prepared with DUB inhibitors present (*Figure 6e–g*). The interaction between Gos1 and COP1/Arf is nearly undetectable if ubiquitination of the components is not preserved. Bet1 interaction with Arf/COPI is also enhanced substantially under conditions that preserve ubiquitination (*Figure 6—figure supplement 1*). Therefore, ubiquitination appears to regulate the assembly and/or stability of the COPI–cargo complex independent of the ubiquitination status of cargo. Not all ubiquitinated SNAREs relied on COPI–ubiquitin interaction for their sorting, For example, Sed5 is ubiquitinated but its localization is not affected by the alterations in the ability of COPI to recognize and bind ubiquitin (*Figure 5h, i*, *Figure 1—figure supplement 1*, Sed5), and Sed5 appears to be independent of COPI for its Golgi localization (*Gao and Banfield, 2020*). A subset of Golgi SNAREs is dependent on the ability of COPI to bind ubiquitin (*Figure 6—figure supplement 1*), and it is likely that other ubiquitin-independent interactions contribute to cargo selection.

One of the long-standing questions about COPI-mediated vesicular trafficking has been the essential roles of α and β′COP WDR domains needed to sustain yeast viability. α and β′COP WDR domains are required for the sorting of dilysine motif COPI cargoes, but cells are viable when all dilysine-binding sites are mutated (*Jackson et al., 2012*). These studies implicated additional roles for the COPI WDR domains in cells. Our data suggest that ubiquitin binding by these COPI WDR domains contributes to the essential function of COPI because yeast harboring β′COP-UBD$_{Doa1}$ grow much better than strains harboring β′COP-Δ2–304 (*Xu et al., 2017*). We have previously shown that the human β′COP-β-propeller also binds polyUb and can function in yeast to support the normal localization of mNG-Snc1, suggesting that the COPI–ubiquitin interaction is functionally conserved. Another key element of vesicle-mediated trafficking is the ability of the coat to bind cargo during vesicle formation, followed by dissociation after the vesicle forms. SNAREs are thought to prime coat assembly through interactions with Arf, ArfGAPs, and COPI as a mechanism to ensure vesicles form with an adequate load of v-SNAREs (*Rein et al., 2002*; *Spang et al., 2010*). The ArfGAP Glo3 contains a BoCCS motif that mediates binding to both COPI and to several different SNAREs, suggesting that Glo3 is a key determinant of the priming complex (*Schindler and Spang, 2007*). However, it is unclear how Glo3 could facilitate coat assembly when its enzymatic function is to inactivate Arf. We have identified a ubiquitin-stabilized complex between Gos1, Arf, and COPI that appears to lack endogenous Glo3 (*Figure 5*). We were able to detect a tagged form of Glo3 in Gos1 immunoprecipitates that lack ubiquitin; however, preserving ubiquitin in these Gos1 pulldowns had no influence on Glo3 recovery even though substantially more Arf and COPI were recovered. The presence of ubiquitin modifications on COPI and Glo3 does not prevent their interaction because we observed enrichment of Glo3 in COPI pulldowns under the same ubiquitin-preserved conditions. Therefore, we suggest that the SNARE/Arf-GTP/COPI priming complex is stabilized by ubiquitination of the components and is devoid of ArfGAP. Arf-GDP and COPI likely dissociate rapidly from the complex as the ArfGAP binds (*Figure 7h*).

Ubiquitin-dependent enrichment of Arf and COPI with SNAREs suggests that cycles of ubiquitination and deubiquitination could control the switch from Arf-GTP/SNARE-mediated assembly of COPI during budding and ArfGAP-mediated disassembly and uncoating of vesicles prior to fusion. Ubiquitination–deubiquitination cycles for key components within the COPI-SNARE system thus may alter the coatomer assembly–disassembly dynamics regulating COPI function.

## Materials and methods
### Reagents
ANTI-FLAG M2 Magnetic Beads (M8823), EZview Red ANTI-FLAG M2 Affinity Gel (F2426), 3xFLAG Peptide (F4799), *N*-ethylmaleimide (E3876), iodoacetamide (GERPN6302), 1,10-phenanthroline (131377), *N*-ethylmaleimide (E3876), deubiquitinase inhibitor PR-619 (SML0430), protease inhibitor tablets (04693159001), and phosphatase inhibitors tablets (PHOSS-RO) were purchased from MilliporeSigma (St Louis, MO). Coomassie Brilliant Blue R-250 Dye (20278) and FM4-64 dye (T-3166) were purchased from Thermo Fisher Scientific (San Jose, CA). ECL Prime Western Blotting Chemiluminescent Substrate (34580) and Pierce Anti-HA Agarose (26181) were purchased from Thermo Scientific

(Rockford, IL). Deubiquitinases (DUBs) Usp2 (E-504), MINDY2 (E-620), MINDY3 (E-621), OTULIN (E-558), AMSH (E-548B), K6-ubiquitin trimer (Ub3) chains (UC-20-025), K11-Ub3 chains (UC-50-025), K29- Ub3 chains (UC-85-025), and K33-Ub3 (UC-105-025) were from BostonBiochem – R&D Systems, Inc (MN, USA).

## Antibodies

ANTI-FLAG antibody produced in mouse (clone M2, F3165, 1:3500) and Anti-HA antibody produced in rabbit (H6908, 1:1000) were purchased from MilliporeSigma (St. Louis, MO). VU101: Anti-ubiquitin Antibody (VU-0101, 1:1000) was purchased from LifeSensors (PA, USA). Anti-mouse HRP conjugate (W4021, 1:10,000) and Anti-Rabbit HRP conjugate (W4011, 1:10,000) were purchased from Promega (Madison, WI). Anti-COPI antibody was a gift from Charles Barlowe (Dartmouth Univ, 1:3000). Anti-Arf antibody (1:3000) used was reported previously (*Liu et al., 2008*). Anti-GST antibody (1:1000) was purchased from Vanderbilt Antibody Product Store (VAPR, Nashville, TN).

## Strains and plasmids

Standard media and techniques for growing and transforming yeast were used. Epitope tagging of yeast genes was performed using a PCR toolbox (*Janke et al., 2004*; *Longtine et al., 1998*). For the construction of mKate-tagged Golgi markers in β'COP-Δ2–304 background, the PXY51 strain was used followed by shuffling of p416-SEC27 with p315-SEC27Δ2–304. The list of yeast strains used in this study is included as a table file (*Supplementary file 1*). Plasmid constructions were performed using standard molecular manipulation. Mutations were introduced using Gibson Assembly Master Mix. The list of plasmids used in this study is included as a table file (*Supplementary file 2*).

## Imaging and image analysis

To visualize mNG- or mScarlet-tagged proteins, cells were grown to early-to-mid-logarithmic phase, harvested, and resuspended in imaging buffer (10 mM $Na_2PHO_4$, 156 mM NaCl, 2 mM $KH_2PO_4$, and 2% glucose). Cells were then mounted on glass slides and observed immediately at room temperature. Images were acquired using a DeltaVision Elite Imaging system equipped with a ×63 objective lens followed by deconvolution using SoftWoRx software (GE Healthcare Life Science). Overlay images were created using the merge channels function of ImageJ software (National Institutes of Health). To quantify SNAREs colocalization, a Pearson's correlation coefficient for the two markers in each cell ($n$ = 3, over 20 cells each) was calculated using the ImageJ plugin Just Another Colocalization Plugin with Costes Automatic Thresholding (*Bolte and Cordelières, 2006*).

Identification and quantitation of fluorescence-based morphological patterns were performed as below: the punctate pattern indicates small, dotted structures, typically around 0.45 µm, the ring-like structures indicate larger, round structures with a hole in the middle, akin to a ring or a donut, roughly 0.6–0.9 µm along the longer and short axis, and elongated tube-like structures indicate tube-like structures whose longer axis is approximately two to five times in length of the shorter axis, approximately 0.6–1.8 µm along the long axis. Each fluorescent structure in the cell was categorized as puncta, ring, or tubule and the number of tubules + ring divided by total fluorescent structures was used to quantify the % tubular and ring structures. Measurements were done in minimum of 50 cells ($n \geq 50$) for three biological replicates. Fluorescence pattern identification and quantitation were repeated in a blinded fashion and/or by a second observer to avoid bias.

## Synthesis of K63 and linear ubiquitin chains

To synthesize K63-linked Ub chains, 2 mM Ub, 300 nM E1, 3 µM UBE2N/UBE2V2 were mixed in the reaction buffer (50 mM Tris–HCl pH 7.5, 50 mM NaCl, 10 mM $MgCl_2$, 20 mM ATP, and 2 mM Dithiothreitol (DTT)) overnight at 37°C. Reactions were quenched by lowering the pH to 4.5 with addition of 5 M ammonium acetate pH 4.4. K63 tri-Ub was isolated and further purified using size-exclusion chromatography (Hiload 26/600 Superdex 75 pg, GE Healthcare) in gel filtration buffer (50 mM Tris–HCl pH 7.5, 150 mM NaCl, 1 mM DTT). Purified chains were buffer exchanged into $H_2O$ and lyophilized. Recombinant M1-linked tri-Ub-FLAG-6XHis was expressed and purified as previously described (*Hepowit et al., 2020*). Briefly, *E. coli* C41 (DE3) cells at $OD_{600}$ of 0.6 were induced with 1 mM Isopropyl ß-D-1-thiogalactopyranoside (IPTG), lysed by sonication in ice-cold Tris buffer (50 mM Tris pH 8.0, 150 mM NaCl, 10 mM imidazole, 2 mM (2-Mercaptoethanol) βME, complete protease

inhibitors (Roche, Basel, Switzerland), 1 µg/ml DNase, 1 µg/ml lysozyme, and 1 mM phenylmethyl-sulfonyl fluoride (PMSF)), and clarified by centrifugation (50,000 × $g$ for 30 min at 4°C) and filtration (0.45 µM filter). M1 tri-Ub was purified to homogeneity by $Ni^{2+}$-NTA affinity column (Thermo Scientific, Rockford, IL) chromatophraphy, HiPrep Q FF anion exchange column (GE Healthcare Life Sciences, Marlborough, MA) chromatography, and HiLoad Superdex size-exclusion column (GE Healthcare Life Sciences, Marlborough, MA) chromatography.

## Construction FLAG-, HA-, and GST-tagged constructs

Multiple strains of *S. cerevisiae* were generated in a manner where one of the components is tagged with an epitope tag. Bet1, Gos1, and Snc1 were C-terminally tagged with 6xHis-TEV-3xFLAG by integration of a PCR product amplified from pJAM617 into the *BET1*, *SNC1*, and *SNC2* locus, respectively (*Janke et al., 2004*). Due to the low recombination rate, a *GOS1* PCR product with longer 5′ and 3′ regions of homology (over 200 bp) was generated from pJAM617 and gene synthesized DNA fragments and integrated into the *GOS1* locus (two-step PCR and integration method). Properly integrated clones were confirmed by genotyping PCR as well as by immunoblot using anti-FLAG antibody. Similarly, COP1, Sec27, and ArfGAP Glo3 were C-terminally tagged with 6xHis-TEV-3xFLAG using two-step PCR and integration method. Additionally, Glo3 was C-terminally tagged with GST (using pFA6a-GST-HisMX6 as a template) in WT *S. cerevisiae* as well as in cells harboring 6xHis-TEV-3xFLAG-tagged Gos1. Efforts to C-terminally tag Ykt6 and Sed5 were unsuccessful; consequently, Ykt6 and Sed5 were N-terminally tagged with 6xHA tag by integration of a PCR product amplified from pYM-N20 cassette (Euroscarf #P30294).

## Purification of FLAG- or HA-tagged proteins

Affinity isolation of FLAG- or HA-tagged proteins was performed with anti-FLAG magnetic beads or Anti-HA Agarose, respectively. 800 $OD_{600}$ of untagged WT cells (BY4742) and cells with C- or N-terminally tagged protein of interest were grown in YPD and harvested by centrifugation when the $OD_{600}$ reached ~0.8. After washing with cold water, the pellets were resuspended in 3 ml lysis buffer (100 mM Tris pH 7.4, 150 mM NaCl, 5 mM ethylenediaminetetraacetic acid (EDTA), 5 mM ethylene glycol-bis(β-aminoethyl ether)-N,N,N′,N′-tetraacetic acid (EGTA), 10% glycerol, 1% Triton X-100, 100 µM PR619, 5 mM 1,10-phenanthroline, 50 mM *N*-ethylmaleimide, phosphatase inhibitors, and complete protease inhibitor tablet). Cells were broken using a Disruptor Genie (Scientific Industries) at 4°C for 10 min at 3000 setting with 0.5 mm diameter glass beads. The lysates were centrifuged at 13,000 rpm for 15 min at 4°C and the supernatant was incubated with 50 µl FLAG or HA beads overnight at 4°C. The next morning the beads were washed 3× with washing buffer (100 mM Tris pH 7.4, 150 mM NaCl, 5 mM EDTA, 1% NP40, 0.5% Triton X-100) and eluted in sodium dodecyl sulfate (SDS) running buffer.

Heterologous expression and purification of GST-β′COP and ubiquitin-binding assays were performed as reported previously (*Xu et al., 2017*). Briefly, 0.5 mM of GST and GST-tagged β′COP (604) proteins immobilized glutathione beads were incubated 250 nM ubiquitin trimer (Ub3) for corresponding linkages, washed 3× and eluted using reduced glutathione.

## DUB treatments

The DUB treatments were performed as described (*Hospenthal et al., 2015*). Briefly, the beads with target proteins were equally split into two parts. One part was subjected to mock treatment as a control, and the other part was incubated with deubiquitinases in the following reaction: 5 µl of 10xDUB reaction buffer (1 M Tris pH 7.4, 1.5 M NaCl, 10 mM DTT), 0.5 µl of deubiquitinase enzyme, and water for a 50 µl reaction volume. The samples were incubated at 37°C for 45 min and reactions were stopped with 2× Laemmli sample buffer by incubating at 95°C for 5 min. Supernatants were collected and used for electrophoresis followed by Western transfer. Deubiquitinases were used following manufacturer recommended concentrations as follows: DUB: the general deubiquitinase Ups2 (1–5 nM); K48-DUB: K48 linkage-specific deubiquitinase MINDY2 and MINDY3 (10–30 nM); K63-DUB: K63 linkage-specific deubiquitinase AMSH (100–500 nM); M1-DUB: OTULIN (0.05 1 µM). Data were generated from independent experiments from three biological replicates and quantified as described later.

For comparative pulldown samples processed under conditions that preserved endogenous ubiquitination (w Ub) or catalyzed removal of ubiquitin (w/o Ub), a similar immunoprecipitation and DUB

protocol were used with the following modifications. The cell pellets (800 $OD_{600}$) were divided into two equal portions. For samples processed under 'w Ub' condition, lysis buffer with deubiquitinase inhibitors (100 mM Tris pH 7.4, 150 mM NaCl, 5 mM EDTA, 5 mM EGTA, 10% glycerol, 0.2% NP40, 100 µM PR619, 5 mM 1,10-phenanthroline, 50 mM N-ethylmaleimide, phosphatase inhibitors, and complete protease inhibitor tablet) was used. Immunoprecipitated samples were washed 2×. As a mock treatment for immunoprecipitated samples under 'w Ub' condition, the deubiquitinase buffer did not have any deubiquitinases. For the samples processed under conditions that catalyzed removal of ubiquitin (w/o Ub), the lysis buffer did not have the deubiquitinase inhibitors 100 µM PR619, 5 mM 1,10-phenanthroline, or 50 mM N-ethylmaleimide, and furthermore the immunoprecipitated samples were processed using DUB buffer containing 1 µl of each deubiquitinase Usp2, AMSH, and OTULIN. Data were generated from independent experiments from three biological replicates and quantified as described later.

For systematic screening of comparative enrichment of Arf with Gos1 under various ubiquitin-preserved/-removed conditions in combination with phosphorylation preserved/phosphorylation removed conditions the samples were processed as described above with the following modifications to the procedure. Cells were lysed in the presence of (1) deubiquitinase inhibitors (PR619, O-PA, and NEM), (2) phosphatase inhibitors (PhosSTOP), (3) both deubiquitinase and phosphatase inhibitors, or (4) no additional inhibitors other than the protease inhibitors. Cell lysis in the presence of deubiquitinase and phosphatase inhibitors is expected to preserve ubiquitin- and phosphorylation-mediated complexes. Cell lysis in the presence of just deubiquitinase or phosphatase inhibitors is expected to preserve only ubiquitin- or phosphorylation-mediated complexes. Cell lysis in the absence of both deubiquitinase and phosphatase inhibitors in expected to not preserve ubiquitin- or phosphorylation-mediated complexes. Following immunoprecipitation using anti-FLAG resin, the samples were treated with (1) deubiquitinases (USP2, AMSH, and Otulin), (2) phosphatases (Lambda phosphatase), (3) both deubiquitinases and phosphatases, and (4) no post-IP treatment. Post-IP deubiquitination (with USP2, AMSH, and Otulin) and/or dephosphorylation (with Lambda phosphatase) is expected to strip off any preserved or remaining ubiquitination and phosphorylation, respectively, from the immunoprecipitated samples.

## Immunoblotting with ECL

Protein samples were separated by 4–20% gradient SDS–polyacrylamide gel electrophoresis followed by immunoblotting. For anti-ubiquitin antibody the membranes were treated with glutaraldehyde solution (supplied with the antibody) as per the manufacturer's protocol and washed with PBS. The membranes were blocked in 5% nonfat milk for 1 hr, incubated with primary antibodies for 3 hr at room temperature, washed five times with Tris-buffered saline (TBS) with 0.1% Tween, incubated with appropriate secondary antibody for 1 hr at room temperature, washed five times and imaged using manufacturer recommended chemiluminescence protocol. The membranes were imaged with AI600 Chemiluminescent Imager (GE Life Sciences). Quantitative analysis of Western blot images was performed using ImageJ software.

## SILAC MS

SILAC-based mass spectrometric analysis of Gos1-FLAG with untagged control was performed using a similar protocol as described previously (*Hepowit et al., 2020*). Briefly, an equal amount of cells (labeled with either light or heavy Arg and Lys) expressing endogenous FLAG-tagged Gos1 or untagged cells were harvested from the mid-log phase and disrupted by bead beating using ice-cold lysis buffer (50 mM Tris–HCl, pH 7.5, 150 mM NaCl, 5 mM EDTA, 0.2% NP-40, 10 mM iodoacetamide, 1 mM 1,10-phenanthroline, 1× EDTA free protease inhibitor cocktail [Roche], 1 mM phenylmethylsulfonyl fluoride, 20 µM MG132, 1× PhosStop [Roche], 10 mM NaF, 20 mM N-[2-hydroxy-3-(1-piperidinyl)propoxy]-3-pyridinecarboximidamide, dihydrochloride (BGP), and 2 mM $Na_3VO_4$). Lysate was clarified by centrifugation at 21,000 × $g$ for 10 min at 4°C and supernatant was transferred into a new tube and diluted with three-fold volume of ice-cold TBS (50 mM Tris–HCl, pH 7.5, 150 mM NaCl). Samples were incubated with 50 µl of EZview anti-FLAG M2 resin slurry (Sigma) for 2 hr at 4°C with rotation. The resin was washed three times with cold TBS and incubated with 90 µl elution buffer (100 mM Tris–HCl, pH 8.0, 1% SDS) at 98°C for 5 min. The collected eluate was reduced with 10 mM DTT, alkylated with 20 mM iodoacetamide, and precipitated with 300 µl

precipitation solution (50% acetone, 49.9% ethanol, and 0.1% acetic acid). Light and heavy protein pellets were dissolved with Urea-Tris solution (8 M urea, 50 mM Tris–HCl, pH 8.0). Heavy and light samples were combined, diluted fourfold with water, and digested with 1 µg MS-grade trypsin (Gold, Promega) by overnight incubation at 37°C. Phosphopeptides were enriched by immobilized metal affinity chromatography using Fe(III)-nitrilotriacetic acid resin as previously described (*MacGurn et al., 2011*) and dissolved in 0.1% trifluoroacetic acid and analyzed by liquid chromatography (LC–MS)/MS using an Orbitrap XL mass spectrometer. Data collected were searched using MaxQuant (ver. 1.6.5.0) and chromatograms were visualized using Skyline (ver. 20.1.0.31, MacCoss Lab). Coimmunoprecipitation followed by SILAC-based mass spectrometric analysis of α-COP-FLAG was performed as described above.

## Statistical analysis

Statistical differences between two groups for SNARE morphology were determined using a Fisher's exact test. For multiple group comparison, one-way analysis of variance on the means using GraphPad Prism (GraphPad Software Inc). Probability values of less than 0.05, 0.01, and 0.001 were used to show statistically significant differences and are represented with *, **, or ***, respectively. To quantify western blot data, at least three independent replicates were used, and intestines were calculated using ImageJ software and statistical analyses, as indicated, were performed using GraphPad Prism.

## Acknowledgements

We thank Dr. Scott Emr (Cornell University) and Dr. Aki Nakano (Riken Institute) for plasmids and strains. We thank Charles Barlow for the anti-COPI antibody. We thank Kristie Lindsey Rose (Vanderbilt University, Proteomics Core Laboratory) for help with processing mass spectrometry samples. Funding sources: These studies were supported by NIH Grants R35GM144123-01 (to TRG), 1R35GM119525 (to LPJ), and R35GM144112 (to JAM). Lauren P Jackson is a Pew Scholar in the Biomedical Sciences, supported by the Pew Charitable Trusts.

## Additional information

### Funding

| Funder | Grant reference number | Author |
| --- | --- | --- |
| National Institutes of Health | R35GM144123-01 | Todd R Graham |
| National Institutes of Health | 1R35GM119525 | Lauren P Jackson |
| National Institutes of Health | R35GM144112 | Jason A MacGurn |
| Pew Charitable Trusts | | Lauren P Jackson |

The funders had no role in study design, data collection, and interpretation, or the decision to submit the work for publication.

### Author contributions

Swapneeta S Date, Conceptualization, Formal analysis, Supervision, Validation, Investigation, Visualization, Methodology, Writing - original draft, Writing - review and editing; Peng Xu, Conceptualization, Formal analysis, Validation, Investigation, Writing - review and editing; Nathaniel L Hepowit, Nicholas S Diab, Formal analysis, Investigation, Writing - review and editing; Jordan Best, Jiale Du, Investigation; Boyang Xie, Investigation, Methodology, Writing - review and editing; Eric R Strieter, Supervision, Investigation, Methodology; Lauren P Jackson, Jason A MacGurn, Conceptualization, Resources, Supervision, Methodology, Writing - review and editing; Todd R Graham, Conceptualization, Resources, Formal analysis, Supervision, Funding acquisition, Investigation, Methodology, Writing - original draft, Writing - review and editing

## Author ORCIDs
Swapneeta S Date (iD) http://orcid.org/0000-0002-4086-110X
Peng Xu (iD) http://orcid.org/0000-0001-7103-3692
Boyang Xie (iD) http://orcid.org/0000-0003-2829-9254
Eric R Strieter (iD) http://orcid.org/0000-0003-3447-3669
Lauren P Jackson (iD) http://orcid.org/0000-0002-3705-6126
Jason A MacGurn (iD) http://orcid.org/0000-0001-5063-259X
Todd R Graham (iD) http://orcid.org/0000-0002-3256-2126

## Decision letter and Author response
Decision letter https://doi.org/10.7554/eLife.80911.sa1
Author response https://doi.org/10.7554/eLife.80911.sa2

## Additional files

### Supplementary files
- MDAR checklist
- Supplementary file 1. Saccharomyces cerevisiae SNAREs used in this study.
- Supplementary file 2. List of strains used in this study.
- Supplementary file 3. List of plasmids used in this study.
- Supplementary file 4. SILAC top 50 hits based on normalized H/L ratio for Gos1.
- Supplementary file 5. SILAC top 50 hits based on normalized H/L ratio for αCOP.
- Source data 1. Source data for *Figures 1, 3–7*.

### Data availability
All data generated or analyzed during this study are included in the manuscript and supporting file; Source Data files have been provided for Figures 1 and 3-7.

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
