## [Editor Report]

This article will be of interest for cell biologists focused on understanding membrane biology, trafficking, and protein ubiquitination, as well as yeast geneticists. The main finding of this paper is that non-degradative ubiquitination is an important mechanism driving COPI-dependent SNARE trafficking and localization.

---

## [Decision Letter]

**Decision letter after peer review:**

[Editors’ note: the authors submitted for reconsideration following the decision after peer review. What follows is the decision letter after the first round of review.]

Thank you for submitting the paper "Ubiquitination drives COPI priming and Golgi SNARE localization" for consideration by *eLife*. Your article has been reviewed by 3 peer reviewers, including Felix Campelo as the Reviewing Editor and Reviewer #1 and the evaluation has been overseen by a Senior Editor.

Comments to the Authors:

We are sorry to say that, after consultation with the reviewers, we have decided that this manuscript in its current form will not be considered further, but we would welcome a new submission if it can address, in full, the reviewers' concerns.

Specifically, all the reviewers valued the strong interest of the scientific question you tackled here. However, after consultation among the three reviewers, we agreed that there are a number of important points that would need to be clarified (requiring a number of additional experimental work), before this manuscript could be further considered. Given that the amount of extra work is relatively considerable (although, we believe, doable in a 3-6 month time period), we decided to reject this manuscript at the present moment, so you can decide whether you want to work on the requested revisions (see below) and – depending on the results found – submit again to *eLife* (as a new submission), or otherwise maybe temper the claims of the paper for a prompt submission elsewhere.

In our opinion, the main points we would like to see so we are convinced about the claims of this paper are:

1) Provide more convincing evidence for a trafficking defect, not just a morphology change. As an example of what can be done (see detailed report below for more details) is to use co-localization as a readout of trafficking. For example, use Mnn9-mCherry or a number of other established Golgi proteins that can be successfully tagged with RFP.

2) Test if the COPI b' mutant (∆2-304) has a (more) general effect on Golgi trafficking and morphology as compared to what they report here. In particular, comparisons to the KK-binding site β'-COP mutant will be an important control to include for most of the experiments, as this is the canonical COPI binding site and is lacking in the β'-COP (∆2-304) mutant.

3) Improve on the image quantitation to better understand the nature of the aberrant compartment where SNAREs relocalize.

There were also other important questions raised by the reviewers (see below) that would require some clarification/explanation (e.g. difference between o/expression end. expression levels; better Bet1 localization together with other Golgi/TGN markers, etc.).

*Reviewer #1 (Recommendations for the authors):*

Date et al. present a thorough study of the effects and roles of protein ubiquitination in the association of different COPI subunits and COPI-associated factors with SNARE proteins in the yeast *S. cerevisiae*. In a previous work from this lab (Xu et al. *eLife* 2017), it was shown that the SNARE Snc1 is mislocalized when COPI is not able to associate with the ubiquitination in Snc1. Here, Date et al. expand on those observations and describe that (i) COPI components, Glo3 (ArfGAP), and some but not all tested Golgi SNARES are ubiquitinated; (ii) the localization of some of these SNARES is dependent on the binding between COPI and the ubiquitinated cargoes (SNAREs) ; and (iii) they propose a model in which non-degradative ubiquitination of SNAREs enhances and/or stabilizes the COPI/Arf/SNARE complex.

I think this is an interesting and timely topic and that the authors have generally provided sufficient experimental support for their claims. I particularly value the clarity of the SILAC data as well as the different ubiquitination/deubiquitination treatments/conditions. However, I think some of the claims could be better supported by having additional experimental data and/or analyses. In particular, the quantification of the fluorescence microscopy images could be improved to better characterize the observed phenotypes. It is not 100% clear to this reviewer whether the observed changes in the morphology of the SNARE-positive structures are a result of a trafficking defect only or that morphological changes of the Golgi cisternae can explain the morphological changes herein reported.

That being said, I think this is a nice piece of work that can potentially be an important contribution to understanding a still obscure cellular process as is the regulation of SNARE incorporation into COPI carriers.

1) Suppl. Data Figure 4: The Aur1 images are of a relatively poor quality (also in S3c). Is there a specific technical reason for that? These data are part of the main ones to support the idea that it is not a change in Golgi morphology but a trafficking defect of the SNAREs that the tubes and rings represent, but I think it will be important to strongly clarify this. Related to this, I have not understood the meaning of the last sentence of this paragraph (line 124-125): what is the evidence that supports the statement that "Bet1 is mislocalizing to a downstream (trans-Golgi) comportment in the COPI mutants tested"? It might be a correct statement, but I missed the evidence to support it, so I would appreciate it if the authors could clarify it.

2) Regarding the quantification of the microscopy images (% tubes and rings), I do appreciate the fact that the authors do it in an unbiased way as possible as detailed in the methods. However, I think this is fundamental for many of the claims in the paper, and it would need to be somehow improved. For instance, the authors present it as a fraction, but also total cell-by-cell distributions of dots, tubes and rings could be informative. Also, I miss details on how the classification was actually done (lines 524-532): how specifically are the different structures identified (software, thresholding, signal to background ratio, etc.), and differentiated (visually? or were the lengths, aspect ratios actually measured?).

3) Line 520: Is PCC the best way to quantify colocalization of these kinds of structures (see e.g. https://www.ncbi.nlm.nih.gov/pmc/articles/PMC3074624/)?

4) Again, I might have missed it, but e.g. in Figure 4f, how do you know that the released mono-ubiquitin (IB:Ub + condition) comes from Ykt6 totally and not also from other ubiquitinated co-IPed proteins? I understand that the blots on the left show indeed that Ykt is ubiquitinated (loss of 100 kDa band), but is there anything else co-IPed there?

*Reviewer #2 (Recommendations for the authors):*

In this manuscript the authors investigate the role of ubiquitination in the trafficking of SNARE proteins. They previously reported that trafficking of the SNARE Snc1 requires ubiquitination. In this work, the authors determine that multiple Golgi-localized SNAREs are also polyubiquitinated. They find that the ubiquitin-binding domain of the COPI β' subunit is required for proper morphology of compartments containing these GFP-SNAREs. They conclude that ubiquitination of these SNAREs is important for their trafficking. Additional co-IP experiments indicate that COPI coat components interact with at least one of these SNAREs in a ubiquitin-dependent manner.

Some of the claims and conclusions may be premature and would benefit from additional experiments, as detailed below:

1. The fluorescence morphology assay used to assess whether SNAREs are being properly trafficked appears problematic. There are at least two issues with this approach: First, the assay cannot distinguish between whether the GFP-SNARE has been trafficked to another compartment or whether the compartment itself has changed morphology. This seems problematic as the model is that ubiquitination is important for SNARE trafficking, yet the assay does not directly measure trafficking. Second, as described in the methods, the authors are scoring the fluorescent structures as either "punctate", "tubules", or "rings". Not only does this appear to involve some judgement in assignments to these categories, but more problematically it results in a loss of information regarding potentially important aspects of the morphology, including numbers of structures per cell, average size or intensity of the structures, etc. Therefore, two genotypes could both be declared to have "wt morphology" based on the % structures that are judged to be punctate, even if in one genotype the cells have many more puncta compared to wt, or if some other parameter is quite different but not measured in the quantitation.

2. My interpretation of the imaging data presented in the manuscript is that the COPI β' mutant (delta2-304) appears to be affecting Golgi trafficking and morphology more generally, rather than specific trafficking of Gos1 and Bet1 Golgi SNAREs. The domain removed from the β' mutant harbors the canonical KKXX COPI cargo binding motif which is known to be important for trafficking of KKXX and KXKXX containing cargos. It is therefore not surprising (and perhaps expected) that Golgi trafficking and morphology would be perturbed by this mutant.

The authors provide some control experiments meant to rule out this possibility, but due to concerns detailed in Point 1 above, and examination of the imaging data examples provided, I do not think the authors have ruled out the possibility that the β' mutant is affecting Golgi morphology rather than specific trafficking of Gos1 and Bet1.

3. The use of overexpression and inducible promoters rather than endogenous expression is potentially problematic. The authors see no difference in results when comparing the CUP1 promoter to the ADH1 promoter, but both promoters could result in non-physiological results if the protein levels are significantly higher than endogenous levels.

4. To summarize this point, there are two main issues with this visual assay: The first is that morphology is simply not a good readout of trafficking. The second is that, as implemented, the quantitation method used does not appear to capture all aspects of morphology, because genotypes that look different are yielding similar quantitative measurements. I will explain here a bit more my observations and reasoning:

The phenotype observed is a morphological change of the compartment that the SNARE localizes to. The authors assert that this means the SNARE now localizes to a different compartment because it cannot engage with COPI to be sorted into COPI vesicles, but an alternative explanation is that the COPI β' mutant, which lacks the ability to bind canonical KKXX cargos, changes the morphology of the compartment that the SNARE normally localizes to. This latter possibility seems supported by the data, as the GFP-SNAREs shown in Figure 1b appears to localize to swollen Golgi compartments in the β' mutant. These swollen compartments also appear to be visible (although I assume to a lesser extent) for the other Golgi SNAREs Sed5 and Tlg1, judging by the images shown in Figure 1-supplement 1a.

The authors try to address this possibility by showing data such as Figure 1-supplement 4 in which they claim that Golgi morphology (as assessed by imaging Sed5, Aur1, and Sec7 fusions) is unaffected by the β' mutant but the images in this figure are not clear and the Sed5 morphology in the wild-type cells appears already a bit unusual in this figure.

Examination of related data in the 2017 *eLife* paper (Figure 4) shows a similar morphological effect in which both the dilysine-binding site mutant and the delta2-304 mutant both result in enlarged Tlg1 compartments.

My concern about the visual phenotype extends to figure 2, because in Figure 2a, GFP-Bet1 appears quite perturbed in the cells with the UBD-Doa1 construct, with an appearance that is not normal for wt cells. In addition, in the 2017 *eLife* paper Figure 4 appears to suggest that the same UBD-Doa1 construct does not rescue the Tlg1 morphological phenotype, which again argues for a more general perturbation of Golgi trafficking. This suggests that the author's use of "% tubular+ring structures" for quantitation may not be the best way to assess whether these SNAREs are properly trafficked.

Therefore, I do not think the authors have ruled out a simple morphological change resulting from aberrant COPI trafficking.

If the authors think the GFP-SNAREs are trafficked to a different compartment in the β' mutant, one alternative would be to demonstrate this via a clear-cut gain and/or loss of colocalization with established compartment markers. The data shown in Figure 1-supplement 3c,d is not convincing.

If they continue using the morphological assay, and are able to significantly improve the quantitation, then additional Golgi membrane proteins that are not SNARE proteins should also be tested in the various mutant conditions, along with the controls already used, in order to better validate the use of this assay to monitor SNARE trafficking.

5. Assuming the authors can improve their methodology and quantitation for assessing whether SNAREs are mis-trafficked, there are still other issues that need to be addressed, mainly centered around the fact that KKXX COPI cargos are also mis-sorted in the β' delta2-304 mutant:

There appears to be some confusion over whether the β' delta2-304 mutant effects KKXX cargos: On lines 175-177, the authors state: "We previously showed that β'-COP Δ2-304 does not perturb Golgi to ER trafficking of cargoes bearing the KKXX or HDEL motifs (Xu et al., 2017). Thus, it is the ability of the β'-COP N-terminal WDR domain to bind ubiquitin, not dilysine motifs, that is critical for SNARE localization." But I think this statement is incorrect, unless I am confused, because in the 2017 Xu et al. *eLife* paper, the authors find that the β' mutant does indeed mislocalize the KXKXX-motif containing cargo Emp47, and they state in that paper: "The β'-COP N-terminal di-lysine binding site has a specific role in sorting Emp47 within the Golgi. As previously reported, β'-COP (∆2-304) and the RKR mutant mislocalizes Emp47 to the vacuole where it is degraded (Eugster et al., 2004). Replacement of the N-terminal propeller of β'-COP with the NZFTab1 or UBDDoa1 domains predictably failed to stabilize Myc-Emp47 because these domains lack the di-lysine binding site (Figure 5C)."

Therefore, I suggest these experiments:

Ideally the authors would have a mutant in β' that blocks ubiquitin binding while preserving the KKXX binding site. In the absence of such a mutant, more experiments and controls are needed to convince me that the model is correct. For example:

Does the KK-binding site mutant of β' result in the same trafficking effects on Gos1 and Bet1?

The DUB fusion experiment shown in Figure 3 should be repeated to include additional controls to examine whether other membrane proteins that localize to the same compartment(s) as Bet1 and Gos1 are similarly affected.

Does a mutant of Gos1 that cannot be ubiquitinated (or is fused to a DUB, as was previously done with Snc1 in the 2017 *eLife* paper) result in the same trafficking defects? [This point arises because the authors state the phenotypes they observe are incomplete because the COPI α subunit also binds Ub. Therefore, a mutant of Gos1 that cannot be ubiquitinated (or is fused to a DUB) would be expected to have stronger sorting phenotypes.]

6. Is there a strong rationale for using the CUP1 promoter instead of native promoters? Even though expression driven by CUP1 is lower than ADH1, it may still be significantly higher than endogenous levels. The authors see no difference in results comparing CUP1 to ADH1 promoters, but both promoters could result in non-physiological results if CUP1 expression is significantly higher than the endogenous levels. Perhaps expression under CUP1 promoter could be compared by western blot to expression by endogenous promoter for the key proteins (Gos1 and Bet1). There also appears to be some cell-to-cell variation arising from varying plasmid copy numbers (and therefore varying expression levels) in cells.

7. Although this may be a relatively minor point, I think it is puzzling that the authors think that Bet1 is localized to the medial/late-Golgi, when it is an established ER-Golgi SNARE (it is the v-SNARE of the Sec22/Bos1/Sed5/Bet1 SNAREpin). There are two potential issues with the experiment they used to determine this (Figure 1-supplement 3a): the first issue is how do the authors know that Aur1 is medial/late? The second issue is that the appearance of Bet1 in the middle row (when co-expressed with Aur1) is quite different from its appearance when co-expressed with Sed5. Furthermore, the bottom row shows a mixture of cells in which some cells express a low level of Bet1 and some cells express a higher level of Bet1. There appears to be some colocalization with Sec7 when Bet1 is expressed at a higher level but not when Bet1 is expressed at a lower level. This highlights potential problems with analyzing proteins expressed on plasmids, as some cells harbor more copies of the plasmid. If the authors wish to convincingly demonstrate the localization of Bet1, the gold-standard is to do time-lapse analysis and use endogenous expression. Although this doesn't seem too relevant to the overall point of the paper, it is nonetheless a poorly supported conclusion.

*Reviewer #3 (Recommendations for the authors):*

In this manuscript Date and colleagues expand on their previous publication (Xu et al., 2017) and show that a sub-set of Golgi SNAREs are mis-localised in a yeast mutant lacking the Β'-COP N-terminal WDR domain. Fusion of a deubiquitinase (DUB) domain to COPI leads to the same mis-localisation phenotype, strengthening their hypothesis. By immunoprecipitating FLAG-tagged SNAREs followed by DUB treatment they are able to show that, while Bet1 does not appear to be ubiquitinated, Gos1 and Snc1 are. SILAC analysis of the Gos1 IP sample shows a stable complex with its partner SNAREs Ykt6 and Sed5 which are also ubiquitinated. By performing IP assays under conditions in which endogenous ubiquitination is preserved or ubiquitin removal is catalyzed, the authors show strong enrichment of Arf and COP1 under ubiquitination conditions. Interestingly this is true also when performing the same experiments with Bet1 which is not itself ubiquitinated. The Arf GAP Glo3 is however not enriched in these SNARE-COPI-Arf complexes and SNAREs localize correctly in a Glo3 mutant. Based on their data, the authors propose a model in which ubiquitination stabilizes an Arf-SNARE-COP complex which does not contain a GAP.

The experiments are generally well executed, however what in my opinion is lacking is a characterization of the aberrant compartments in which the SNAREs are mis-localized in the BetaCOP mutant. This would greatly help to understand the mechanism. If SNAREs are not recycled back via COPI do they leak to downstream compartments? The authors say that Bet1 is mis-localizing to a downstream (trans-Golgi) compartment (lines 126-127), however, Figure 1 figure supplement 3c-d show a decreased co-localisation with the late Golgi marker Aur1. Is it possible Bet1 is in the TGN? Co-localization with Sec7 is not tested. Snc1 is an exocytic SNARE however it mis-localises to the same aberrant compartment as the other Golgi SNAREs?

---

## [Author Response]

[Editors’ note: the authors resubmitted a revised version of the paper for consideration. What follows is the authors’ response to the first round of review.]

In our opinion, the main points we would like to see so we are convinced about the claims of this paper are:1) Provide more convincing evidence for a trafficking defect, not just a morphology change. As an example of what can be done (see detailed report below for more details) is to use co-localization as a readout of trafficking. For example, use Mnn9-mCherry or a number of other established Golgi proteins that can be successfully tagged with RFP.

Our experiments show that a subset of SNAREs localize to tube-like and ring-like structures when the ubiquitin-binding domain in β’COP is deleted (β’COP Δ2-304 cells). To further characterize these aberrant structures, we performed colocalization analysis of mNG-Gos1 with mKate tagged early Golgi markers Anp1 and Mnn9, late Golgi markers Sec7 and Chs5 as well as with FM4-64 to mark endosomes. Our extensive colocalization experiments show significant loss of colocalization of mNG-tagged Gos1 with both early-Golgi as well as both late-Golgi markers in β’COP Δ2-304 cells compared to wild-type (WT) control (Figure 2, Figure 2 figure supplements 1). We showed in the original submission a reduction in Bet1 co-localization with the Golgi marker Aur1. None of the Golgi markers tested displayed the same morphological phenotype observed with Bet1, Gos1 or Snc2. Altogether these data indicate a significant loss of SNAREs from Golgi compartments in β’COP Δ2-304. We further tested if Gos1 was being mislocalized to endosomes by labeling cells with FM4-64 for a short period of time to stain endosomes (Figure 2, Figure 2 figure supplements 4). While some of the tubular structures stained weakly with FM4-64, no significant increase in colocalization of mNG-Gos1 or mNG-Bet1 with FM4-64 was observed in β’COP Δ2-304 relative to WT cells; therefore, these Golgi SNAREs are not being mislocalized to the endosomal system. Our conclusion for these studies as stated on page 8 is that “We suspect that a failure to retrieve these SNAREs during maturation of the Golgi cisternae, combined with a normal retrieval of most resident Golgi proteins, results in Gos1 and Bet1 accumulating in dead-end compartments that we describe here as sequestration compartments.” Additional figures with these colocalization studies are now in a new Figure 2 and Figure 2 —figure supplement 1 and 4 and described on page 7 and 8.

2) Test if the COPI b' mutant (∆2-304) has a (more) general effect on Golgi trafficking and morphology as compared to what they report here. In particular, comparisons to the KK-binding site β'-COP mutant will be an important control to include for most of the experiments, as this is the canonical COPI binding site and is lacking in the β'-COP (∆2-304) mutant.

To further address this comment, we compared the localization of mNG-Gos1 and mNG-Bet1 in the β’COP-RKR mutant (which is not capable of binding dilysine-motif cargo) with WT cells and with β’COP Δ2-304 cells. No observable difference in the appearance and localization of mNG-Bet1 and mNG-Gos1 was observed in β’COP-RKR mutant compared to WT cells. This result is now shown in (Figure 2 figure supplement 3b).

Data from our previous work and current efforts indicate that there is no general effect on Golgi morphology or overall vesicle-trafficking events in β'-COP (∆2-304) as supported by the following observations:

2.1: Localization of Gos1, Bet1, Snc1 and Snc2 is restored to the WT pattern when the terminal βpropeller domain of β’COP is replaced with ubiquitin-binding domain of Doa1 (Figure 3a-3f). While Doa1UBD can bind ubiquitin, it is not capable of binding dilysine cargo (Xu et al. 2017). If the mislocalization of SNARE was the result of impaired ability of β’COP to bind dilysine cargo, restoration to WT phenotype in Doa1-UBD fusion constructs would not have been possible. We previously demonstrated that the Doa1-UBD fusion failed to rescue the Emp47 trafficking defect exhibited by ∆2-304 (Xu et al. 2017). Thus, di-lysine binding and ubiquitin-binding are separable functions of the COPI β’ subunit. Emp47 is the only cargo we are aware of that has been shown to be mislocalized in the β’COP RKR mutant. Most dilysine cargoes are handled efficiently by alphaCOP.

2.2: We now show that the localization and appearance of several Golgi markers including Anp1, Mnn9, Aur1, Sec7, Chs5 and Sed5 are not different in β’COP Δ2-304 cells compared to WT cells (Figure 2 and its supplements). We previously showed that Rer1 trafficking, a sensitive indicator for overall loss of COPI function, was unaffected in β’COP Δ2-304 cells. These data indicate that there is no general defect in Golgi morphology or COPI-dependent trafficking.

2.3: The localization and appearance of several ER, Golgi, and endosomal SNAREs are not affected in β’COP Δ2-304 cells compared to WT cells (Figure 1 figure supplement 1), indicating this phenomenon is specific to a subset of SNAREs and not a generalized trafficking defect.

2.4: In data not presented here, we initially looked at the trafficking of several different GFP-tagged proteins (Mup1, Tat2, Pma1, Ena1, Can1, Ina1, Pdr5) known or suspected to recycle between the plasma membrane, endosomes, Golgi and back to the plasma membrane because we hypothesized that the COPI-Ub interaction was specifically driving this pathway (similarly to Snc1). However, none of these proteins were mislocalized in β’COP Δ2-304 cells.

3) Improve on the image quantitation to better understand the nature of the aberrant compartment where SNAREs relocalize.

We have now included additional figures with colocalization analysis of mNG-SNAREs with Golgi markers and endosomes. We have use Mander’s co-efficients to quantify the degree of co-localization for the new co-localization data. We have also provided detailed scoring and quantitative analyses for (a) the number of fluorescent mNG-Gos1 and mNG-Bet1 structures in WT and in β’COP Δ2-304 cells and (b) the average size of these structures as requested by reviewer 1 (Figure 2 and its supplements).

There were also other important questions raised by the reviewers (see below) that would require some clarification/explanation (e.g. difference between o/expression end. expression levels; better Bet1 localization together with other Golgi/TGN markers, etc.).

Please see point-by-point responses below.

Reviewer #1 (Recommendations for the authors):Date et al. present a thorough study of the effects and roles of protein ubiquitination in the association of different COPI subunits and COPI-associated factors with SNARE proteins in the yeast *S. cerevisiae*. In a previous work from this lab (Xu et al. eLife 2017), it was shown that the SNARE Snc1 is mislocalized when COPI is not able to associate with the ubiquitination in Snc1. Here, Date et al. expand on those observations and describe that (i) COPI components, Glo3 (ArfGAP), and some but not all tested Golgi SNARES are ubiquitinated; (ii) the localization of some of these SNARES is dependent on the binding between COPI and the ubiquitinated cargoes (SNAREs) ; and (iii) they propose a model in which non-degradative ubiquitination of SNAREs enhances and/or stabilizes the COPI/Arf/SNARE complex.I think this is an interesting and timely topic and that the authors have generally provided sufficient experimental support for their claims. I particularly value the clarity of the SILAC data as well as the different ubiquitination/deubiquitination treatments/conditions. However, I think some of the claims could be better supported by having additional experimental data and/or analyses. In particular, the quantification of the fluorescence microscopy images could be improved to better characterize the observed phenotypes. It is not 100% clear to this reviewer whether the observed changes in the morphology of the SNARE-positive structures are a result of a trafficking defect only or that morphological changes of the Golgi cisternae can explain the morphological changes herein reported.That being said, I think this is a nice piece of work that can potentially be an important contribution to understanding a still obscure cellular process as is the regulation of SNARE incorporation into COPI carriers.

We thank this reviewer for the supportive comments and we have worked to better address the trafficking and morphological phenotypes as describe above.

1) Suppl. Data Figure 4: The Aur1 images are of a relatively poor quality (also in S3c). Is there a specific technical reason for that? These data are part of the main ones to support the idea that it is not a change in Golgi morphology but a trafficking defect of the SNAREs that the tubes and rings represent, but I think it will be important to strongly clarify this. Related to this, I have not understood the meaning of the last sentence of this paragraph (line 124-125): what is the evidence that supports the statement that "Bet1 is mislocalizing to a downstream (trans-Golgi) comportment in the COPI mutants tested"? It might be a correct statement, but I missed the evidence to support it, so I would appreciate it if the authors could clarify it.

We have now performed SNARE colocalization analysis with several additional Golgi markers (two early Golgi and two late Golgi markers) that are mKate-tagged and give a better observable fluorescence pattern. We have also analyzed cells pulsed with FM4-64 to label the endosomes and find that Gos1 or Bet1 are not substantially mislocalized to endosomes. The new analyses indicate that in β’COP Δ2-304 cells there is a significant loss of SNAREs from early, medial, and late Golgi compartments to what we are now describing as sequestration compartments (Figure 2 and its supplements). We, at this point, have not identified another non-SNARE marker protein that co-labels this compartment. However, the colocalization analyses, complementation analyses with general ubiquitin-binding domains, and comparisons of Golgi morphology in WT vs β’COP Δ2-304 show that SNAREs are lost from Golgi compartments without affecting Golgi morphology or general trafficking. The text has been revised to clarify the nature of the compartment we are observing as described above for the editor’s summary.

2) Regarding the quantification of the microscopy images (% tubes and rings), I do appreciate the fact that the authors do it in an unbiased way as possible as detailed in the methods. However, I think this is fundamental for many of the claims in the paper, and it would need to be somehow improved. For instance, the authors present it as a fraction, but also total cell-by-cell distributions of dots, tubes and rings could be informative. Also, I miss details on how the classification was actually done (lines 524-532): how specifically are the different structures identified (software, thresholding, signal to background ratio, etc.), and differentiated (visually? or were the lengths, aspect ratios actually measured?).

As outlined above, colocalization studies with 4 mKate-tagged Golgi markers and FM4-64 marked early endosomes were performed to provide unbiased, software-based readouts for changes in colocalization using Mander’s coefficients on thresholded images. The rings and tubes were visually scored and binned in a blinded fashion for most of the analyses. We have now included detailed information about scoring of these structures and additional plots as requested in the new Figure 2 and its supplementary figures. We find no significant change in the total number of fluorescent structures for Bet1 and Gos1, but do observe a significant change in the distribution of rings/tubes versus punctae between WT vs β’COP Δ2-304.

3) Line 520: Is PCC the best way to quantify colocalization of these kinds of structures (see e.g. https://www.ncbi.nlm.nih.gov/pmc/articles/PMC3074624/)?

This is a good point and we have quantified all of the new data using Mander’s coefficients.

4) Again, I might have missed it, but e.g. in Figure 4f, how do you know that the released mono-ubiquitin (IB:Ub + condition) comes from Ykt6 totally and not also from other ubiquitinated co-IPed proteins? I understand that the blots on the left show indeed that Ykt is ubiquitinated (loss of 100 kDa band), but is there anything else co-IPed there?

We agree with the reviewer’s suggestion that there is a possibility that the ubiquitin we are detecting may also come from other proteins co-IPed with Ykt6. We have carefully used the language that ubiquitination is associated with SNAREs and/or IPed complexes to try and avoid the possible misinterpretation that ubiquitin is solely linked with the protein being IPed. With that being noted, the smeared patterns detected on the western blots (or average mass) associated with ubiquitination is different when different SNAREs and COPI components are IPed and probed for ubiquitin. If we were observing the same complex over and over again with different components being pulled down, we anticipate the smeared pattern would be identical. These differences in the ubiquitin-associated migration pattern and different extent of ubiquitin being released suggests that the bait protein is the primary contributor for the associated ubiquitination. In support of this contention, COPI co-IPs with Bet1 and yet we cannot detect the presence of ubiquitin associated with this complex above background, even though we know many COPI subunits are ubiquitinated.

Reviewer #2 (Recommendations for the authors):In this manuscript the authors investigate the role of ubiquitination in the trafficking of SNARE proteins. They previously reported that trafficking of the SNARE Snc1 requires ubiquitination. In this work, the authors determine that multiple Golgi-localized SNAREs are also polyubiquitinated. They find that the ubiquitin-binding domain of the COPI β' subunit is required for proper morphology of compartments containing these GFP-SNAREs. They conclude that ubiquitination of these SNAREs is important for their trafficking. Additional co-IP experiments indicate that COPI coat components interact with at least one of these SNAREs in a ubiquitin-dependent manner.Some of the claims and conclusions may be premature and would benefit from additional experiments, as detailed below:1. The fluorescence morphology assay used to assess whether SNAREs are being properly trafficked appears problematic. There are at least two issues with this approach: First, the assay cannot distinguish between whether the GFP-SNARE has been trafficked to another compartment or whether the compartment itself has changed morphology. This seems problematic as the model is that ubiquitination is important for SNARE trafficking, yet the assay does not directly measure trafficking. Second, as described in the methods, the authors are scoring the fluorescent structures as either "punctate", "tubules", or "rings". Not only does this appear to involve some judgement in assignments to these categories, but more problematically it results in a loss of information regarding potentially important aspects of the morphology, including numbers of structures per cell, average size or intensity of the structures, etc. Therefore, two genotypes could both be declared to have "wt morphology" based on the % structures that are judged to be punctate, even if in one genotype the cells have many more puncta compared to wt, or if some other parameter is quite different but not measured in the quantitation.

Please see responses above. We have now addressed these concerns with additional experiments presented as the new Figure 2 and associated supplemental figures.

2. My interpretation of the imaging data presented in the manuscript is that the COPI β' mutant (delta2-304) appears to be affecting Golgi trafficking and morphology more generally, rather than specific trafficking of Gos1 and Bet1 Golgi SNAREs. The domain removed from the β' mutant harbors the canonical KKXX COPI cargo binding motif which is known to be important for trafficking of KKXX and KXKXX containing cargos. It is therefore not surprising (and perhaps expected) that Golgi trafficking and morphology would be perturbed by this mutant.

Please see the response to the editor. We have revised the manuscript in several places to better explain how published data and our current results make it highly unlikely that our results are a secondary effect of perturbing dilysine trafficking.

The authors provide some control experiments meant to rule out this possibility, but due to concerns detailed in Point 1 above, and examination of the imaging data examples provided, I do not think the authors have ruled out the possibility that the β' mutant is affecting Golgi morphology rather than specific trafficking of Gos1 and Bet1.

Please see responses above (comment 1 and 2). We have now imaged a large number of Golgi markers and quantified the number and size of the Golgi compartment marked by Anp1, Mnn9, Chs5 and Sec7 (new Figure 2). Gos1 and Bet1 quantifiably colocalize with Golgi markers in WT cells but this colocalization is significantly lost in the COPI mutant. We hoped to find a Golgi or endosomal marker that labeled the abnormal structures but this is not what we have found. However, we feel that the improved analysis and quantitation makes a strong case that Bet1 and Gos1 are mislocalized from the compartments where they normally reside.

3. The use of overexpression and inducible promoters rather than endogenous expression is potentially problematic. The authors see no difference in results when comparing the CUP1 promoter to the ADH1 promoter, but both promoters could result in non-physiological results if the protein levels are significantly higher than endogenous levels.

Please see response above to reviewer 1.

4. To summarize this point, there are two main issues with this visual assay: The first is that morphology is simply not a good readout of trafficking. The second is that, as implemented, the quantitation method used does not appear to capture all aspects of morphology, because genotypes that look different are yielding similar quantitative measurements. I will explain here a bit more my observations and reasoning:The phenotype observed is a morphological change of the compartment that the SNARE localizes to. The authors assert that this means the SNARE now localizes to a different compartment because it cannot engage with COPI to be sorted into COPI vesicles, but an alternative explanation is that the COPI β' mutant, which lacks the ability to bind canonical KKXX cargos, changes the morphology of the compartment that the SNARE normally localizes to. This latter possibility seems supported by the data, as the GFP-SNAREs shown in Figure 1b appears to localize to swollen Golgi compartments in the β' mutant. These swollen compartments also appear to be visible (although I assume to a lesser extent) for the other Golgi SNAREs Sed5 and Tlg1, judging by the images shown in Figure 1-supplement 1a.

We’ve looked through several Sed5 images, including the Figure 1 supplement 1 images and do not see any difference between WT and mutant. We thank the reviewer for correctly pointing out that Tlg1 is displaying tubular structures and we inadvertently failed to mention this in the initial submission. We assumed these were the same structures that also label with Snc1 based on our earlier studies and decided to focus on other Golgi SNAREs for this paper. We have revised the text to indicate we also observe abnormal structures with Tlg1.

The authors try to address this possibility by showing data such as Figure 1-supplement 4 in which they claim that Golgi morphology (as assessed by imaging Sed5, Aur1, and Sec7 fusions) is unaffected by the β' mutant but the images in this figure are not clear and the Sed5 morphology in the wild-type cells appears already a bit unusual in this figure.

See comments above and the new Figure 2. We haven’t performed a lot of imaging experiments with Sed5 but this mScarlet-Sed5 morphology did not strike us as being unusual. The mNG-Sed5 images in Figure 1 supplement 1 showed a larger number of spots per cell than mScarlet-Sed5, which we assume may be because of the brighter fluorescence of mNG.

**Author response image 1. sa2fig1:** 

Examination of related data in the 2017 eLife paper (Figure 4) shows a similar morphological effect in which both the dilysine-binding site mutant and the delta2-304 mutant both result in enlarged Tlg1 compartments.

This is a good point and we have revised the manuscript accordingly.

My concern about the visual phenotype extends to figure 2, because in Figure 2a, GFP-Bet1 appears quite perturbed in the cells with the UBD-Doa1 construct, with an appearance that is not normal for wt cells. In addition, in the 2017 eLife paper Figure 4 appears to suggest that the same UBD-Doa1 construct does not rescue the Tlg1 morphological phenotype, which again argues for a more general perturbation of Golgi trafficking. This suggests that the author's use of "% tubular+ring structures" for quantitation may not be the best way to assess whether these SNAREs are properly trafficked.

The morphological influence of the UBD-Doa1 construct on Tlg1 morphology has not been quantified in the 2017 *eLife* paper or the current work. Extensive quantitation across multiple biological replicates would be needed to draw conclusions about Tlg1 and we feel that initiating a careful study of Tlg1 is beyond the scope of the current work. Please see above for the detailed discussion regarding Golgi morphology.

Therefore, I do not think the authors have ruled out a simple morphological change resulting from aberrant COPI trafficking.If the authors think the GFP-SNAREs are trafficked to a different compartment in the β' mutant, one alternative would be to demonstrate this via a clear-cut gain and/or loss of colocalization with established compartment markers. The data shown in Figure 1-supplement 3c,d is not convincing.

We have now provided a clear-cut loss of colocalization with 4 Golgi markers.

If they continue using the morphological assay, and are able to significantly improve the quantitation, then additional Golgi membrane proteins that are not SNARE proteins should also be tested in the various mutant conditions, along with the controls already used, in order to better validate the use of this assay to monitor SNARE trafficking.

This has now been done using Golgi glycosyltransferases and Sec7 as markers.

5. Assuming the authors can improve their methodology and quantitation for assessing whether SNAREs are mis-trafficked, there are still other issues that need to be addressed, mainly centered around the fact that KKXX COPI cargos are also mis-sorted in the β' delta2-304 mutant:

Based on our careful analysis of literature and our own experiments, we have not identified any dilysine cargo other than Emp47 that is mislocalized by any mutations to β’ COP that renders it incapable of binding dilysine cargo. If we have missed any published work, please provide a reference. There seems to be a common view in the trafficking field that sorting of dilysine cargos is the essential function of COPI and this emphasizes why the current study is important. Here is a quote from Jackson et al. 2012 (PMID 23177648).

“Finally, we generated a yeast strain in which both the α- and β'-COP N-terminal WD-repeat domains had lost the ability to bind the carboxy terminus of dilysine motifs (*sec27::URA3* R15A K17A R59A *ret1::TRP1* R13A K15S R57S). Although viable at 37°C, the ability of this mutant to traffic both KKxx and KxKxx reporter constructs was severely impaired (Figure S4E). The inability to support retrograde dilysine-based transport causes a slight growth defect at 37°C, resulting in smaller colonies as compared to wild-type, but this mutant is not lethal.”

There appears to be some confusion over whether the β' delta2-304 mutant effects KKXX cargos: On lines 175-177, the authors state: "We previously showed that β'-COP Δ2-304 does not perturb Golgi to ER trafficking of cargoes bearing the KKXX or HDEL motifs (Xu et al., 2017). Thus, it is the ability of the β'-COP N-terminal WDR domain to bind ubiquitin, not dilysine motifs, that is critical for SNARE localization." But I think this statement is incorrect, unless I am confused, because in the 2017 Xu et al. eLife paper, the authors find that the β' mutant does indeed mislocalize the KXKXX-motif containing cargo Emp47, and they state in that paper: "The β'-COP N-terminal di-lysine binding site has a specific role in sorting Emp47 within the Golgi. As previously reported, β'-COP (∆2-304) and the RKR mutant mislocalizes Emp47 to the vacuole where it is degraded (Eugster et al., 2004). Replacement of the N-terminal propeller of β'-COP with the NZFTab1 or UBDDoa1 domains predictably failed to stabilize Myc-Emp47 because these domains lack the di-lysine binding site (Figure 5C)."

The reviewer is correct that the second sentence should not have started with “Thus” because that conclusion is based on more data than the preceding sentence spells out. Emp47 has a variant dilysine motif that depends on its interaction with β'-COP and is the only dilysine cargo we are aware of that exclusively relies on β’COP. Emp47 is mislocalized to vacuole in β'-COP ∆2-304 as well as in the β'-COP RKR mutant. In case of SNAREs, they do not have a dilysine binding motif and SNARE mislocalization is restored when terminal propeller of β'-COP is replaced with an unrelated ubiquitin-binding domain UBDDoa1 (which can bind ubiquitin but not dilysine cargo). This supports our observation that SNARE interaction with COPI is not dilysine motif dependent but it is dependent on the ability of β'-COP to bind ubiquitin. We have modified this sentence to improve accuracy.

Therefore, I suggest these experiments:Ideally the authors would have a mutant in β' that blocks ubiquitin binding while preserving the KKXX binding site. In the absence of such a mutant, more experiments and controls are needed to convince me that the model is correct. For example:Does the KK-binding site mutant of β' result in the same trafficking effects on Gos1 and Bet1?

Additional experiments were done to analyze the appearance and localization of mNG-Bet1 and mNGGos1 in β'-COP RKR mutant (see above). No mislocalization or change of morphology for mNG-Gos1 or mNG-Bet1 was observed in β'-COP RKR mutant.

The DUB fusion experiment shown in Figure 3 should be repeated to include additional controls to examine whether other membrane proteins that localize to the same compartment(s) as Bet1 and Gos1 are similarly affected.

The original COPI-DUB fusions have a relatively weak phenotype. We now have a better collection of DUB fusions to other COPI subunits that are providing fascinating new data (we believe we can now interfere with α-COPI binding to ubiquitin). We are performing several studies with multiple cargoes and controls to provide a full characterization of these new COPI-DUB fusion constructs. We hope the reviewer will give us the latitude to report these additional experiments in the next manuscript on this topic. We feel these COPI-DUB fusion studies are moving beyond the scope of the current work and that we have adequately addressed the primary concerns related to compartment morphologies and marker localizations with the new data we have included (please see above).

Does a mutant of Gos1 that cannot be ubiquitinated (or is fused to a DUB, as was previously done with Snc1 in the 2017 eLife paper) result in the same trafficking defects? [This point arises because the authors state the phenotypes they observe are incomplete because the COPI α subunit also binds Ub. Therefore, a mutant of Gos1 that cannot be ubiquitinated (or is fused to a DUB) would be expected to have stronger sorting phenotypes.]

In many cases, ubiquitination is promiscuous, meaning if the primary site of ubiquitination is mutated another Lys within the same substrate is often ubiquitinated. Additionally, there are several lysines within Gos1 and mutating all of these lysines could lead to non-ubiquitin-dependent but Lys-dependent phenotypes. Further, mNG has several lysine residues, which could potentially be ubiquitinated in the context of the fusion. Due to these reasons the Lys mutagenesis studies have not been employed. Two people in our lab tried to fuse DUB to mNG-Gos1, but the construct seems to be toxic to *E. coli* because all of the clones isolated carried deletions or mutations.

6. Is there a strong rationale for using the CUP1 promoter instead of native promoters? Even though expression driven by CUP1 is lower than ADH1, it may still be significantly higher than endogenous levels. The authors see no difference in results comparing CUP1 to ADH1 promoters, but both promoters could result in non-physiological results if CUP1 expression is significantly higher than the endogenous levels. Perhaps expression under CUP1 promoter could be compared by western blot to expression by endogenous promoter for the key proteins (Gos1 and Bet1). There also appears to be some cell-to-cell variation arising from varying plasmid copy numbers (and therefore varying expression levels) in cells.

We compared expression of Snc1 and Bet1 from the CUP1 and ADH promoters by western blot and correlated this to the endogenous expression of ADH and yeast SNAREs (from the SGD page – average values from several studies). We see more than a 100-fold reduction in SNARE expression from the CUP1 promoter under the conditions we use to induce expression, which puts this in the range of normal expression for the yeast SNAREs. These data were published as supplemental figure 3 in Best et al. (2020) and this was done concurrently with many of the imaging studies for the submitted manuscript. This was described in the submitted manuscript but we have added another sentence in the Results section to better communicate this result.

7. Although this may be a relatively minor point, I think it is puzzling that the authors think that Bet1 is localized to the medial/late-Golgi, when it is an established ER-Golgi SNARE (it is the v-SNARE of the Sec22/Bos1/Sed5/Bet1 SNAREpin). There are two potential issues with the experiment they used to determine this (Figure 1-supplement 3a): the first issue is how do the authors know that Aur1 is medial/late? The second issue is that the appearance of Bet1 in the middle row (when co-expressed with Aur1) is quite different from its appearance when co-expressed with Sed5. Furthermore, the bottom row shows a mixture of cells in which some cells express a low level of Bet1 and some cells express a higher level of Bet1. There appears to be some colocalization with Sec7 when Bet1 is expressed at a higher level but not when Bet1 is expressed at a lower level. This highlights potential problems with analyzing proteins expressed on plasmids, as some cells harbor more copies of the plasmid. If the authors wish to convincingly demonstrate the localization of Bet1, the gold-standard is to do time-lapse analysis and use endogenous expression. Although this doesn't seem too relevant to the overall point of the paper, it is nonetheless a poorly supported conclusion.

It is true that Bet1 is an established ER/Golgi SNAREpin and it is puzzling that Bet1 localizes to the Golgi and does not appear to recycle back to the ER. Dieter Schmitt’s lab explored the steady-state distribution of Bet1 using a C-terminal tag and found that it localized to the Golgi by immunofluorescence and subcellular fractionation (Ossipov et al. 1999). The tag contained an alphafactor fragment that was cleaved rapidly by Kex2 after the fusion protein was synthesized, indicating that Bet1 moves rapidly to the TGN. They also saw that Bet1 remained in the Golgi in a sec23 (COPII) ts mutant at the nonpermissive temperature. We thought this was a surprising result but we repeated their experiments using the mNG-N-terminal tag and obtained the same results. In addition to Aur1, we have now co-localized mNG-Bet1 with several more Golgi markers and find it more enriched with lateGolgi markers than early Golgi markers in WT cells. To account for potential cell-cell variability in the plasmid-driven expression, we have ensured to include a large number of cells across biological replicates in our analyses. The same modest variation in expression is observed in WT and mutant cells and so this shouldn’t impact the overall conclusion. The primary point of these studies is not to define the precise Golgi compartment where Bet1 localizes or its dynamic retrieval in WT cells and we feel the time-lapse studies are beyond the scope of the current study. In addition, we deleted a paragraph in the Discussion on the implications of Bet1 trafficking on its SNARE function at the ER/Golgi interface. We will incorporate these ideas into a review article where they can be developed more completely.

Reviewer #3 (Recommendations for the authors):In this manuscript Date and colleagues expand on their previous publication (Xu et al., 2017) and show that a sub-set of Golgi SNAREs are mis-localised in a yeast mutant lacking the Β'-COP N-terminal WDR domain. Fusion of a deubiquitinase (DUB) domain to COPI leads to the same mis-localisation phenotype, strengthening their hypothesis. By immunoprecipitating FLAG-tagged SNAREs followed by DUB treatment they are able to show that, while Bet1 does not appear to be ubiquitinated, Gos1 and Snc1 are. SILAC analysis of the Gos1 IP sample shows a stable complex with its partner SNAREs Ykt6 and Sed5 which are also ubiquitinated. By performing IP assays under conditions in which endogenous ubiquitination is preserved or ubiquitin removal is catalyzed, the authors show strong enrichment of Arf and COP1 under ubiquitination conditions. Interestingly this is true also when performing the same experiments with Bet1 which is not itself ubiquitinated. The Arf GAP Glo3 is however not enriched in these SNARE-COPI-Arf complexes and SNAREs localize correctly in a Glo3 mutant. Based on their data, the authors propose a model in which ubiquitination stabilizes an Arf-SNARE-COP complex which does not contain a GAP.The experiments are generally well executed, however what in my opinion is lacking is a characterization of the aberrant compartments in which the SNAREs are mis-localized in the BetaCOP mutant. This would greatly help to understand the mechanism. If SNAREs are not recycled back via COPI do they leak to downstream compartments? The authors say that Bet1 is mis-localizing to a downstream (trans-Golgi) compartment (lines 126-127), however, Figure 1 figure supplement 3c-d show a decreased co-localisation with the late Golgi marker Aur1. Is it possible Bet1 is in the TGN? Co-localization with Sec7 is not tested. Snc1 is an exocytic SNARE however it mis-localises to the same aberrant compartment as the other Golgi SNAREs?

Please see the comments above and the new Figure 2 along with its supplemental figures.